# Evolutionary history biases inferences of ecology and environment from $\delta^{13}$C but not $\delta^{18}$O values

Kirsty M. Edgar [1], Pincelli M. Hull[2] & Thomas H.G. Ezard [3,4]

Closely related taxa are, on average, more similar in terms of their physiology, morphology and ecology than distantly related ones. How this biological similarity affects geochemical signals, and their interpretations, has yet to be tested in an explicitly evolutionary framework. Here we compile and analyze planktonic foraminiferal size-specific stable carbon and oxygen isotope values ($\delta^{13}$C and $\delta^{18}$O, respectively) spanning the last 107 million years. After controlling for dominant drivers of size-$\delta^{13}$C and size-$\delta^{18}$O trends, such as geological preservation, presence of algal photosymbionts, and global environmental changes, we identify that shared evolutionary history has shaped the evolution of species-specific vital effects in $\delta^{13}$C, but not in $\delta^{18}$O. Our results lay the groundwork for using a phylogenetic approach to correct species $\delta^{13}$C vital effects through time, thereby reducing systematic biases in interpretations of long-term $\delta^{13}$C records—a key measure of holistic organismal biology and of the global carbon cycle.

[1] School of Geography, Earth and Environmental Sciences, University of Birmingham, Birmingham B15 2TT, UK. [2] Department of Geology & Geophysics, Yale University, PO 208109, New Haven, CT 06520-8109, USA. [3] Biological Sciences, University of Southampton, Life Sciences Building 85, Highfield Campus, Southampton SO17 1BJ, UK. [4] Ocean and Earth Science, National Oceanography Centre Southampton, University of Southampton Waterfront Campus, Southampton SO14 3ZH, UK. Correspondence and requests for materials should be addressed to K.M.E. (email: k.m.edgar@bham.ac.uk)

In the ocean, the stable isotopic and trace element composition of a single protistan clade, the Foraminifera, provides the most comprehensive means of tracking the dynamic marine environment, and therefore global climate, over long time scales[1]. Foraminifera live in the plankton and the benthos (planktonic and benthic foraminifera, respectively) and many build tests (i.e., shells) out of calcium carbonate[2]. These calcareous tests preserve geochemical traces of the environment in which they precipitated and, when preserved in deep-sea sediments, record the temporal dynamics of the ocean and climate system[1, 3]. Since the first analyses of stable carbon and oxygen isotopes[4] ($\delta^{13}C$ and $\delta^{18}O$, respectively), one of the greatest uncertainties in directly interpreting isotopic records as environmental and/or climatic signals has been the impact of the biology of the organism, e.g., metabolism, calcification, growth rate etc. on these isotopic signals[4, 5].

Most foraminiferal species have distinct offsets between the geochemical composition of their tests and the ambient environment known collectively as vital effects[5–7]. The impact of certain aspects of biology and ecology on foraminiferal test geochemistry are relatively well-constrained, including the presence or absence of dinoflagellate photosymbionts and the effect of body size, but many others such as metabolism are not[5, 8]. To extract primary environmental signals from foraminiferal geochemical records, we need to quantify how each biotic vital effect transforms the abiotic signal of interest. Here we test whether the evolutionary relatedness of taxa impacts differences in $\delta^{13}C$ and $\delta^{18}O$ values amongst species (e.g., ref. [6, 9]).

Paleoceanographers and geochemists know that vital effects are important and they account for them when generating stable isotope records by controlling for symbiont effects and depth habitat, and establishing isotopic offsets between species. Closely related species within the same genus are typically chosen for long-term environmental reconstructions, across extinction boundaries or when moving between low- and high-latitude assemblages. When closely related taxa are not available, researchers switch to other clades on the basis of similarities in symbiont ecology, depth habitat and relative isotopic differences. Assuming that genera are defined meaningfully, this practice should minimize the effect of accumulated differences in isotopic fractionation amongst lineages when jumping between species through time. Geochemists and paleoceanographers therefore use an implicit evolutionary hypothesis when developing their sampling strategy, but have not tested it formally. We seek to move beyond this ad-hoc incorporation of vital effect dependence to an explicitly evolutionary setting embedding changes among species and correlating biological effects in mathematical models of evolutionary divergence. This approach has the benefit of providing quantitative information on the influence (and potential bias) of evolutionary history on geochemistry by specifically addressing what degree all taxa present in any given time interval may be evolutionarily offset from the environment. Quantifying this offset facilitates an ability to reset calibrations (relative to the environment) across extinction and/or biogeographic boundaries.

There are ample hints that foraminiferal isotopic signatures might contain a strong imprint of evolutionary history. Closely related foraminifera often share multiple morphological and ecological characteristics[10]. For example, extant shallow dwelling, dinoflagellate-bearing planktonic foraminiferal species all occur in one branch of the phylogeny[10]. Thus, one might expect two species from the same genus to have more similar vital effects than two species from different genera driven by symbiont and light-dependent effects. In addition, shallow water or photosymbiotic species might have additional metabolic adaptations to their unique life history (i.e., their life cycle), further offsetting their isotopic composition from environmental values

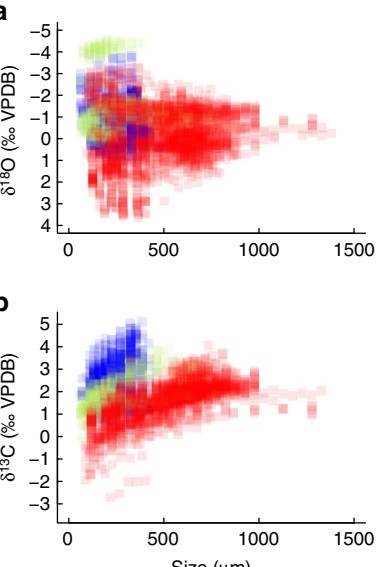

**Fig. 1** Planktonic foraminiferal body size and stable isotope values over the past 107 million years. Multispecies compilation of size-specific oxygen (**a**) and carbon (**b**) isotope trends (Supplementary Data 1). The data are color coded for visualization purposes only; Quaternary-Neogene–Oligocene (red), Paleocene–Eocene (blue), and the Cretaceous (green)

beyond the direct effects of symbionts and depth habitat. However, it's important to note that species from two different genera may be more similar to each other if they are evolutionarily more closely related (in terms of time or number of speciation events since their last common ancestor) than two more distantly related species within the same genus. In other words, it may be the evolutionary similarity of taxa that provides the most information about their geochemical similarity. Thus, phylogenetic evolutionary relatedness provides more resolution than current best practices in testing and controlling for evolutionary history on geochemical interpretations. The incorporation of phylogenetic comparative methods[11] into paleoceanographic approaches could directly impact the interpretation of records of long-term and abrupt climatic change because turnover in species compositions often accompany these environmental events.

Here we build and analyze a new compilation of $\delta^{13}C$ and $\delta^{18}O$ data from planktonic foraminiferal species to quantify the impact of evolution on planktonic foraminiferal species-specific $\delta^{13}C$ and $\delta^{18}O$ values during the Cenozoic Era (0–66 million years). Phylogenetic comparative methods acknowledge explicitly that species are not independent data points in statistical analysis[12] by using shared evolutionary history as a backbone constraining trait diversification[12, 13], ideally given a particular evolutionary model[11, 14]. Nevertheless, such comparative evolutionary thinking has yet to be applied to understand the apparent "black box" of species-specific vital effects in stable isotope and/or trace element variation. To provide a strict test of the relative utility of a phylogenetic framework for interpreting geochemical records, we first account for as much isotopic variance as possible using a suite of environmental, biological, and preservational factors that have been shown as important controls on isotopic expression using a much larger data set spanning the last 107 million years. We then test the otherwise unexplained offsets for an evolutionary signal using a time-calibrated, species-level phylogenetic tree[15] and an explicit model of how traits evolve. In this way, we provide an estimate of the effect of evolutionary history on the $\delta^{13}C$ and $\delta^{18}O$ of planktonic foraminifera. The methods and hypotheses tested

**Table 1 Improvement in model likelihood with increasing model complexity**

| | Oxygen | | | | Carbon | | | |
|---|---|---|---|---|---|---|---|---|
| | df | logLik | AIC | ΔAIC | df | logLik | AIC | ΔAIC |
| Model 1: No environmental predictors but differences among species' means | 5 | −4239 | 8488 | 2925.7 | 5 | −2104.2 | 4218.5 | 1773.2 |
| Model 2: Environmental predictors but no differences among species' means | 50 | −4544.1 | 9188.2 | 3626 | 50 | −2698 | 5496 | 3050.7 |
| Model 3: Environmental predictors and species' mean differences | 50 | −3861.3 | 7822.6 | 2260.3 | 50 | −1699 | 3497.9 | 1052.6 |
| Model 4: Environmental predictors and species' mean and variation differences | 154 | −2635.6 | 5579.3 | 17 | 154 | −1073.3 | 2454.6 | 9.3 |
| Model 5: Simplified environmental predictors and species' mean and variation differences | 137 | −2644.1 | 5562.3 | 0 | 137 | −1085.6 | 2445.3 | 0 |

Successive inclusion of mean differences among species (random effects), environmental + mean species differences, and environmental + mean species differences + heteroscedastic (i.e., different amounts of variation per species) species differences improves model fit substantially. The impact of model simplification is relatively minor. The Akaike Information Criterion (AIC) is a measure of model fit and is a compromise between variance explained and parameters used. Burnham and Anderson (p. 70 of ref. [65]) consider models with a difference in AIC scores (ΔAIC) of < 2 to be "essentially equivalent" and ΔAIC > 10 implies that support for the lesser model is " essentially none"

in this study provide a foundation for future development of more refined models to disentangle the coevolution of life and the planet.

## Results

**Predictive models of foraminiferal size-isotope trends**. Our compilation of sieve size-specific $\delta^{13}C$ and $\delta^{18}O$ data for Cenozoic and Cretaceous planktonic foraminifera encompasses 105 species and 3797 data points spanning the past 107 million years and all major ocean basins (Supplementary Data 1; Methods). The sieve size-specific data were used because a positive body size-$\delta^{13}C$ trend is the primary means of identifying dinoflagellate-bearing photosymbiotic taxa in the fossil record. A minimum of three data points were required for each species and site[5, 16]. Each major time interval (upper Cretaceous (107–66 million years ago), early Cenozoic (66–34 million years ago), and late Cenozoic (34–0 million years ago)) includes representative species of the major planktonic foraminiferal life-history strategies, including mixed layer symbiont-bearing species, mixed layer asymbiotic species, and thermocline dwelling species. Although we discuss major patterns in the data in terms of geological time, it is important to note that time was not explicitly incorporated in the analyses.

The raw data (Fig. 1) show a clear separation between samples from the Cretaceous and early Cenozoic 'greenhouse' climate (before Antarctic glaciation at 34 Ma) and those from the younger 'icehouse' climate state (< 34 Ma). Older samples have smaller maximum body sizes (< 450 μm), and relatively high $\delta^{13}C$ and low $\delta^{18}O$ values by comparison to the latter-half of Cenozoic (Fig. 1). $\delta^{18}O$ values show a large spread at small body sizes that reduces with increasing body size. This pattern is consistent with the higher abundance of both smaller species and body sizes within populations compared to larger taxa, and with the largest species occurring only at (sub)tropical latitudes where high temperatures drive the relatively low and consistent $\delta^{18}O$ values.

Non-linear mixed models[17] were used to test the relative importance of four environmental (ocean basin, biome, and background climate represented by benthic foraminiferal $\delta^{13}C$ and $\delta^{18}O$ values), four biological (depth habitat, algal symbiont type, spinosity, wall structure), and two preservational (water depth of core, carbonate preservation state) factors hypothesized to drive the $\delta^{13}C$ and $\delta^{18}O$ variation in Fig. 1 (for full details see Methods). All analyses were performed in R[18] (Supplementary Notes 1–6). Note that the effects of changes in the global ocean and climate across the past 107 million years are accounted for by the variable "background climate" determined from a global compilation of time-specific benthic foraminiferal $\delta^{13}C$ and $\delta^{18}O$

values[1, 19]. Models took the form

$$y_i = a_i + bx + cx^2 + \varepsilon_i \qquad i = 1, 2, \ldots, 105, \qquad (1)$$

where $y$ is the isotope of interest ($\delta^{13}C$ or $\delta^{18}O$); $x$ is the size class of foraminifers analyzed; and the regression parameters, which describe the size-dependent change in isotopic composition include $a$ (the intercept), $b$ (the linear trend) and $c$ (the non-linear component of the data). $\varepsilon_i$ is the residual error structure. The importance of our 10-predictors (environmental, biological and preservation factors) were examined for all regression parameters ($a$, $b$, and $c$) as fixed effects. In other words, the effect of the 10-predictors were assumed to be the same across all species in the analysis. Species-specific vital effects (i.e., random effects in statistical terminology) were only applied to the intercept parameter (hence the subscript on $a$) and, importantly, $\varepsilon_i$, due to data limitations (Supplementary Note 2, Supplementary Fig. 1). This effectively imposes the strong assumption that species-specific vital effect differences are manifested in just the intercept parameter $a$. Future analysis with much larger or more balanced isotopic data sets may demonstrate that species-specific vital effects are even more pervasive, and affect the slope and saturation parameters as well—we could not test these possibilities, given the limitations of the current data. In our current analysis, the species-specific random effects could represent biological factors like metabolism, beyond the fixed effects directly tested.

The importance of species-specificity (i.e., random effects) for both $\delta^{13}C$ and $\delta^{18}O$ is clear from two simplified models (Table 1). Model 1, which includes size dependence and species' mean differences via random effects, overwhelmingly outperforms Model 2, which contains environmental predictors but no differences among species (i.e., no random effects). This second model features the 10 environmental, biological, and preservational predictors, which can all influence each of the size-dependent parameters ($a$, $b$, and $c$ in Eq. (1)). Increasing model complexity to include both predictors (i.e., fixed effects) and species mean differences (i.e., random effects) improves the model performance further as evidenced by the decreasing Akaike Information Criteria (AIC) values in Table 1—a measure of model fit. However, the most dramatic model improvement is observed with the additional inclusion of heteroscedastic variance, i.e., differential amounts of variation among species, in Model 4. This additional complexity allows some species to exhibit substantial isotopic variation, while others can be quite constrained[20]. Finally, we generated a minimum adequate model (MAM; Supplementary Notes 2 and 3) using backward model

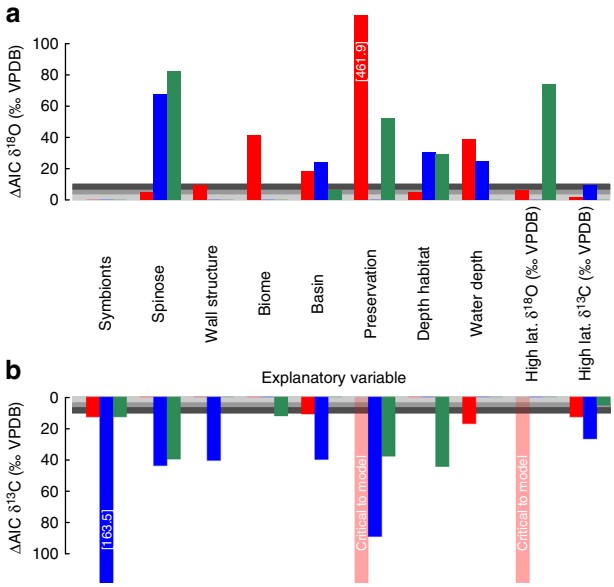

**Fig. 2** Testing the controls on size-specific stable isotope values. Each colored set of bars represents the contribution of each of the environmental, biological and preservational variables to the intercept (red), linear slope (blue) or non-linear slope (green) of the model fitted to the raw $\delta^{18}O$ (**a**) or $\delta^{13}C$ (**b**) data. The height of each bar is the difference in the Akaike Information Criterion (ΔAIC) between MAM outputs with and without the listed focal variable, an indication of the explanatory power of each variable. Horizontal grey bars follow the criteria of ref. [65]. The light grey zone (ΔAIC < 4) indicates similar model outputs with and without the focal variable, i.e., low explanatory power of variable. The grey bar (ΔAIC 4–7), dark grey bar (ΔAIC 7–10) and the white region above (ΔAIC > 10) indicate increasingly divergent amounts of support between model outputs, and thus the focal variable explains substantial variation. Translucent red bars marked 'critical to model' indicates that the model fails to converge on a solution if it is removed

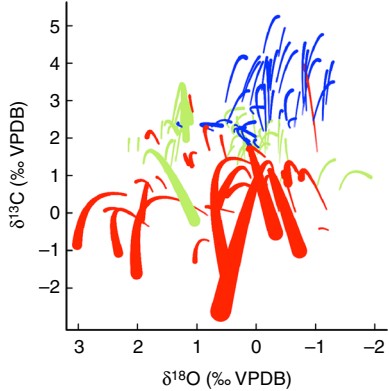

**Fig. 3** Synthetic predictions of species $\delta^{13}C$ and $\delta^{18}O$ size trends. Model-averaged predictions are made for each of the species in our data set, assuming a common oceanographic setting (i.e., subtropical Atlantic Ocean sediment sample at 1500 m water depth). Lines are colored for visualization purposes only as follows: Quaternary-Neogene–Oligocene (red), Paleocene–Eocene (blue), and Cretaceous (green). Predictions are only made across the range of body sizes available for each species in the original data set (which is capped at 1150 μm in this projection) and the line thickness is directly proportional to the species body size

simplification[17, 21], but the improvements in performance from this final step were much more modest (Table 1, Model 5).

During model simplification, we observed that preservation accounted for the greatest drop in model performance as each term was removed in turn from the $\delta^{18}O$ MAM (ΔAIC = 461.9; e.g., Fig. 2—the higher the bar the greater the explanatory power of the predictor). The major predictor of the $\delta^{13}C$ MAM was symbiont influence on the linear slope (ΔAIC = 163.5), followed by preservation. The $\delta^{13}C$ MAM failed to converge without preservation or benthic foraminiferal $\delta^{18}O$, a tracer of global temperatures and ice volume over time[1], on regression intercepts. One interpretation of this convergence failure is that, it implies that the climatic factors traced by benthic foraminiferal $\delta^{18}O$ and carbonate preservation play a pivotal role in shaping the magnitude of species offsets in $\delta^{13}C$ and the overall non-linear environmental trend.

Our approach captures the variation in size-specific trends through time and for each species relatively well (Supplementary Note 2, Supplementary Fig. 6 and Supplementary Note 3, Supplementary Fig. 10). We therefore use this approach to generate model-averaged predictions of species-specific size-isotope trends to enable direct comparison of size-isotope trends and species-specific offsets among all species assuming a uniform environmental state in a single synthetic ocean core with "Excellent" (i.e., "glassy" sensu[22]) preservation at 1500 m water depth in the subtropical Atlantic (Fig. 3). More specifically, as the MAM is only one of many similarly performing models, all model predictions were made using a model–averaging approach that

weights the influence of each model by its Akaike weight. The Akaike weight can be interpreted as the probability that the particular model is the correct one of those being compared. As such, all models (not just the MAM) feed into the reported species-specific size-isotope predictions but those with an AIC close to the minimum obtained (i.e., the best model) are weighted more heavily in predictions than those with weaker explanatory power.

Resulting model-averaged predictions (Fig. 3) reveal a distinct temporal separation of species size-$\delta^{13}C$ trends and a much wider range of species-specific offsets in $\delta^{13}C$ than in $\delta^{18}O$ (~4.5 vs. 3.1) when visualized by geologic time interval. As time was not incorporated explicitly in the analysis, this apparent clustering in geologic time emerges indirectly even after accounting for the explanatory variables (e.g., long-term background climate trends) in Fig. 2. Exploring the differences among the species projections in Fig. 3 reveals that specific clades are clumped within our isotopic space: e.g., Truncorotaloididae, which includes *Acarinina*, *Igorina*, *Morozovella*, *Morozovelloides* and *Praemur-ica*, cluster tightly, as do *Subbotina*, and to a lesser extent, the Neogene *Globigerinoides* (Fig. 4). Major clades have distinct occurrences in geologic time –potentially accounting for the temporal structure observed in Fig. 3.

**Phylogenetic controls on stable isotope offsets.** The potential role of shared evolutionary history in the above anecdotes suggests non-independence among species, violating the fundamental statistical assumption of independence among the data points. The consequences for isotope geochemistry are biased environmental or ecological reconstructions using stable isotope values, which could potentially propagate through to future climate projections used to test or 'tune' climate or Earth System models. The key role of shared evolutionary history (i.e., phylogenetic dependence) on organismal traits, and the need to control for it in analyses, is now routine in evolutionary biology[11, 12]. However, it is not currently used in geochemistry, paleoceanography, or stable isotope ecology, despite its potential to refine paleoclimate reconstructions across multispecies records. Here we introduce this evolutionary toolkit to isotope geochemistry to test if shared evolutionary history has a pervasive effect on geochemical signals, building on the visual anecdotal

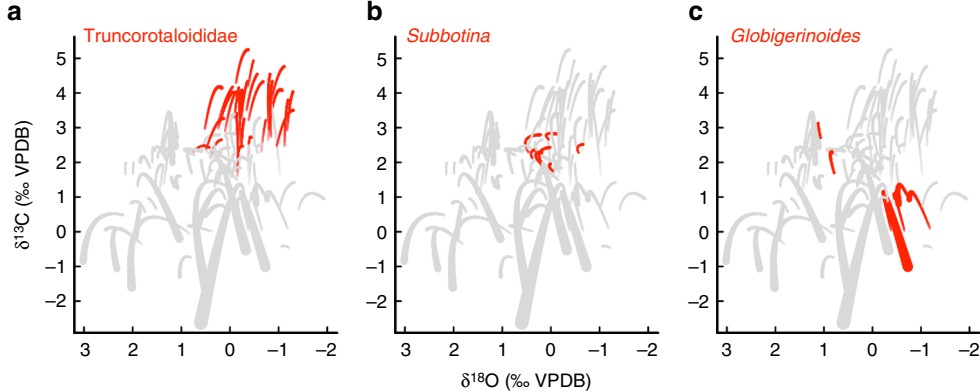

**Fig. 4** Synthetic predictions of δ18O and δ13C size trends for key clades. All species shown in grey as in Fig. 3, but here with common mixed layer, photosymbiont-bearing groups from the Paleogene (**a**) and Neogene (**c**), and asymbiotic thermocline dweller *Subbotina* (**b**) picked out in red to highlight 'clumping' of closely related taxa in isotopic space

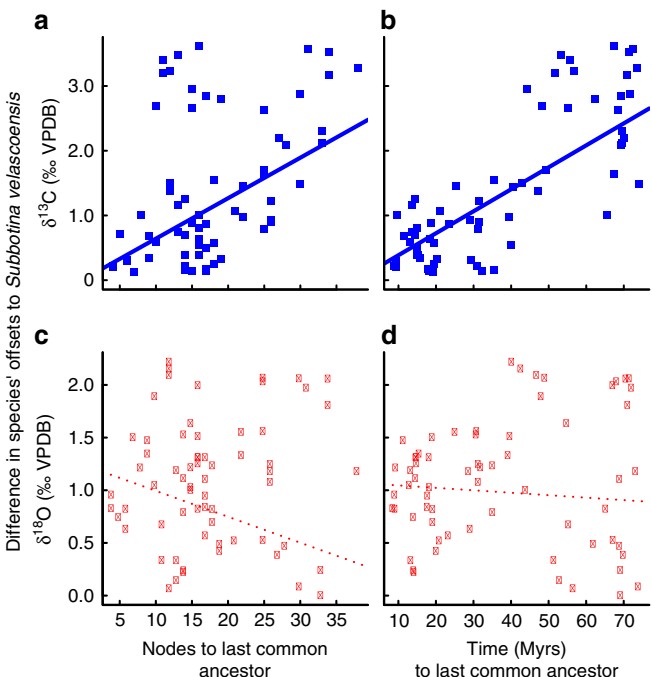

**Fig. 5** Testing the role of evolutionary history on *Subbotina velascoensis* δ13C and δ18O offsets. The evolutionary distance of *S. velascoensis* to the last common ancestor with each of the other 65 species modeled here is calculated using two different approaches; number of nodes (i.e., splitting events) in (**a**) and (**c**) and time in millions of years, (**b**) and (**d**). Linear regressions highlight a significant positive trend in δ13C, indicating a strong relationship between evolutionary distance and species offsets and a lack of a significant trend, and thus relationship to evolutionary history in δ18O

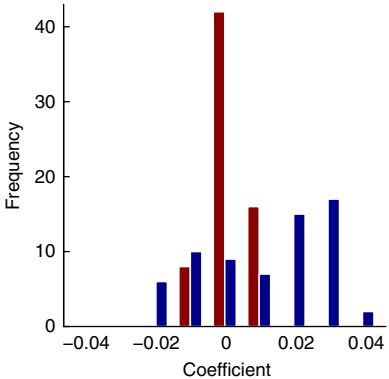

**Fig. 6** Relationship between evolutionary distance and interspecific δ18O and δ13C offsets. We repeated the process outlined in Fig. 5 for each of the other 65 Cenozoic macroperforate species in the data set. Subsequently the co-efficient of the regression slope between the evolutionary distance and each of the model-averaged species-specific δ18O and δ13C offsets relative to the other 65 taxa in the data set are presented here. Blue bars are for δ13C and red bars are for δ18O. Note that regression slopes are more positive for δ13C than for δ18O, indicating that closely related species are likely to have more similar isotopic offsets than more distantly related ones supporting a role for evolutionary history in shaping δ13C but not δ18O offsets

evidence shown in Figs. 3 and 4. We restrict this phylogenetic analysis to the macroperforate clade of the Cenozoic Era, for which we have a recent and complete species-level taxonomy[15].

In phylogenetic comparative analysis, the similarity between species depends upon the evolutionary distance to their last common ancestor[13]. We illustrate how this effect works using the Paleocene taxa *Subbotina velascoensis* as a case study, which is located centrally in our model-averaged projected isotope space (Fig. 4). Figure 5 shows the model-averaged isotopic variation between offsets in *S. velascoensis* and each of the other 65 species in our data set as a function of evolutionary distance. Species

closely related to *S. velascoensis* fall on the left-hand side of the plots; species more distantly related to *S. velascoensis* fall on the right-hand side of the plots. We find a strongly positive relationship between the species' difference in δ13C offsets from *S. velascoensis* and their evolutionary distance from *S. velascoensis*, whether measured as number of speciations or time to last common ancestor (Fig. 5a, b). In this case study, this relationship is not replicated for the corresponding δ18O offsets (Fig. 5c, d). This means species closely related to *S. velascoensis* typically have very similar δ13C offsets, but may have a δ18O offset across the full Cenozoic range.

The trends shown in the *S. velascoensis* case study (Fig. 5) are representative of our full data set. We repeated the analysis on each of the macroperforate species in our Cenozoic data set, hypothesizing positive correlations between evolutionary distance and the size of the difference in δ13C or δ18O offsets (i.e., the random effects) if evolutionary relationships account for some of the variance missed by the model fixed effects (i.e., those factors shown in Fig. 2). If shared evolutionary history 'matters' for

geochemical interpretations, then closely related species should have similar offsets and distantly related species should have more dissimilar offsets. When we examined the relationship between evolutionary distance and $\delta^{18}O$ offset differences amongst all species, we failed to reject the null hypothesis that regression coefficients were equally likely to be positive or negative (sign test for nodes [39 of 66, $p > 0.05$] and time (41 of 66, $p > 0.05$)). In contrast, and as we found for *S. velascoensis*, we did reject the null hypothesis for the relationship between $\delta^{13}C$ offset differences and evolutionary distance (sign test for nodes [43 of 66, $p < 0.05$]; for time [45 of 66, $p < 0.01$]). Furthermore, a paired Wilcox rank-sum test rejected the null hypothesis of no difference between $\delta^{18}O$ and $\delta^{13}C$ offsets ($V = 1480$, $p < 0.05$ for nodes; $V = 1834$, $p < 0.001$ for time) and supported the result that regression slopes between evolutionary distance and $\delta^{13}C$ offset differences are more positive in $\delta^{13}C$ than $\delta^{18}O$ (Fig. 6). Together, these tests indicate that evolutionary distance is related to the similarity of $\delta^{13}C$ offsets amongst species, even after multiple environmental, preservational, and biological factors are accounted for.

While graphical displays (Figs. 5 and 6) illuminate the importance of time and divergence on isotopic state, embedding such systematic biases in phylogenetic approaches and formal evolutionary models is preferable for inferring how they emerged in evolutionary history[23]. We use two different analytical approaches (Pagel's $\lambda$[24], and Blomberg's $K$[25]) to test if shared evolutionary history could explain the patterns in our species-specific, model-averaged $\delta^{18}O$ and $\delta^{13}C$ random effects across the whole phylogeny. These two approaches are superior to those detailed in the previous paragraph as they simultaneously assess relationships across all species on the entire phylogeny rather than independently testing each. For both $\lambda$ and $K$, a value of 1 implies that trait evolution across the phylogeny is consistent with a Brownian motion model, whereas a value of 0 indicates no evolutionary signal, i.e., all species are assumed to be equally related to all other species. A value between 0 and 1 indicates that trait evolution is slower than expected in a Brownian motion model[24, 25]. The maximum likelihood estimate of Pagel's $\lambda$ for $\delta^{18}O$ was 0 with a 95% confidence interval (CI) of 0–0.41, indicating that $\delta^{18}O$ is equivalently explainable by the assumption that evolutionary history doesn't matter (i.e., all species are equally related) as by the assumed evolutionary history of the current phylogeny. By contrast, the maximum likelihood estimate of $\lambda$ for $\delta^{13}C$ was 0.71 with 95% CI of 0.42–0.92. These $\lambda$ statistics control for symbiont status, biome and vertical depth habitat, which also carry an evolutionary signal[10]. In other words, the evolutionary signal may be diminished by including these factors as fixed effects. To test for this, we ran an alternative model estimating random effects when evolutionarily influenced fixed effects (symbiont status, biome and vertical depth habitat) were excluded. These revised random effects were highly similar with regards to $\lambda$ ($\delta^{18}O$ unchanged at 0 [95% CI: 0–0.48]; $\delta^{13}C$ increased to 0.79 (95% CI: 0.49–0.97)) as our original, more conservative analysis. Blomberg's $K$ supports these results: $K$ was 0.28 for $\delta^{18}O$ and 0.94 for $\delta^{13}C$. While the 95% CI for $\delta^{13}C$ $\lambda$ doesn't quite include 1, when considered together with the imbalance in the raw data and the statistical procedures used to construct the synthetic column (all data and R code in the Supplementary Notes 1–5), it is difficult to confidently discount the suggestion that the evolution of species-specific offsets in $\delta^{13}C$ cannot be explained by the Brownian motion model. Regardless of whether the confidence interval bounds on $\lambda$ include 1 or not, the $\delta^{13}C$ $\lambda$-estimate very strongly supports a role for shared evolutionary history in shaping the distribution of $\delta^{13}C$ across the phylogeny, in stark contrast to the situation for $\delta^{18}O$.

## Discussion

Although "nothing makes sense in biology except in the light of evolution"[26], this is the first phylogenetic comparative test for the influence of shared evolutionary history on geochemical signals. Over the past 107 million years, there have been systematic changes in the size-specific $\delta^{13}C$ and $\delta^{18}O$ of planktonic foraminifera (Fig. 1). Non-linear mixed effects modeling captures this variation well (Supplementary Notes 2–3) and highlights the importance of four principal factors in determining the $\delta^{13}C$ and $\delta^{18}O$ of planktonic foraminiferal tests: fossil preservation, benthic foraminiferal $\delta^{18}O$ (a proxy for global climate), symbiont status, and additional species-specific effects. What these additional species-specific effects are is unknown, but one possibility is that they relate to metabolic differences amongst species. Many of the factors we found to be important in Fig. 2 are well-known to the paleoceanographic community as key determinants of foraminiferal isotopic composition[3, 5, 27]. Our results emphasize the relative importance of each factor. Model performance improved with the successive inclusion of environmental and biological complexity (Table 1). The huge decreases in AIC scores with species-specific random effects and differential isotopic variation amongst species is consistent with a large amount of isotopic variation being left unexplained by the environmental, biological, and preservational factors typically considered in paleoceanographic studies (e.g., Fig. 2).

Intriguingly, one of the key environmental predictors of planktonic $\delta^{13}C$ values appears to be benthic foraminiferal $\delta^{18}O$, a proxy for changes in high-latitude temperature and global ice volume (Fig. 2). Over the focal interval (0–107 million years ago), benthic foraminiferal $\delta^{18}O$ tracks global cooling from the Cretaceous greenhouse to the modern day icehouse characterized by bipolar continental ice sheets[1, 19]. In contrast, benthic foraminiferal $\delta^{13}C$, a proxy of long-term changes in the global ocean dissolved inorganic carbon (DIC) pool, has a relatively minor influence. The relative importance of benthic foraminiferal $\delta^{18}O$ implies that it must co-vary with some other factor that primarily influences planktonic foraminiferal $\delta^{13}C$ alone. There are a number of possibilities for altering the $\delta^{13}C$ composition of planktonic foraminifera in greenhouse relative to icehouse climate states, including: temperature-dependent planktonic foraminiferal metabolism altering their $\delta^{13}C$ relative to surrounding waters; temperature-dependent organic-matter remineralization in the upper ocean resulting in altered $\delta^{13}C$ depth-profiles of the DIC; increased primary production (due to temperature-dependent metabolic rates) elevating surface ocean $\delta^{13}C$ due to greater depletion of surface ocean DIC pools[9] and finally, varying planktonic foraminiferal associations, e.g., with bacterial symbionts under different climate modes[28]. In summary, we infer that metabolic processes modified by both evolutionary and environmental factors dominate $\delta^{13}C$ values.

In order to test for an evolutionary effect on $\delta^{18}O$ and $\delta^{13}C$, we applied several common tests for the influence of evolution on the species-specific random effects generated by the non-linear mixed effects model. The random effects can be thought of as the isotopic variance specific to each species after accounting for the environmental, preservational, and biological factors affecting all species in a similar way (i.e., those fixed effects factors shown in Fig. 2). Our evolutionary tests might be considered conservative tests of the importance of evolution for geochemical interpretations because many of the fixed effects (e.g., symbiont status) have an evolutionary signal (although re-running analyses without these controls did not change the results qualitatively, Supplementary Note 6). We choose to perform this conservative test, which effectively asks 'what can evolution tell us in addition to what we already know?' to see if the exclusion of this perspective in current geochemical and paleoceanographic studies might be

systematically biasing interpretations. We found that the random effects for $\delta^{13}C$ were indeed influenced by evolution, with more closely related taxa having more similar offsets (Figs 4 and 5) as tested by Pagel's $\lambda$ and Blomberg's K. This was not true of $\delta^{18}O$. $\delta^{18}O$ showed no significant evolutionary influence on species offsets, regardless of the evolutionary test used. Given the impact of shared evolutionary history on species-specific $\delta^{13}C$ offsets and the environmental influence on $\delta^{13}C$ discussed above, it is unclear to what extent planktonic foraminiferal $\delta^{13}C$ values accurately record the DIC of ambient ocean waters at any point in time in the geological record.

The lack of evolutionary signal in species-specific $\delta^{18}O$ effects is consistent with our understanding that foraminiferal $\delta^{13}C$ values are more directly impacted by biology (e.g., symbionts), and $\delta^{18}O$ by environment (e.g., depth habitat). Returning to the raw isotopic data, $\delta^{13}C$ data appear distinctly separated by time interval (Fig. 1). This separation in isotope space is also apparent in model predictions (Fig. 3), with species at the limits of the projected isotopic range corresponding to a cohesive clade of Paleogene taxa (Fig. 4a). To reiterate, even after controlling for the environment, organismal biology, and preservation in our model predictions, there is still a systematic variation in $\delta^{13}C$ values through time related to evolution in planktonic foraminifera. This implies that the holistic 'biology' of foraminifera has changed through time, even for those species with similar inferred ecologies. The evolution of $\delta^{13}C$ offsets across the phylogeny has the scope to impact long-term $\delta^{13}C$ records and interpretations of the marine carbon cycle. However, as demonstrated here, evolutionary distance (i.e., relatedness and particularly time to last common ancestor) does provide a promising means of quantifying and controlling for this bias.

While our model-averaged species predictions could potentially be a used as a predictive tool to correct existing paleoceanographic records (and this remains a long-term goal of this work), we caution against broad generalizations at this point because of the structural dependences (i.e., limitations) in the data compilation illustrated below. For instance, *Menardella menardii* spans the largest range of body sizes in our compilation and this is reflected in the model-averaged $\delta^{13}C$ and $\delta^{18}O$ predictions (see longest and widest red line in Fig. 3). The raw data from any given oceanographic location does not show the same curvature for *M. menardii* as we project, but the error between observation and projection is equally balanced around 0 (Supplementary Note 2, Supplementary Fig.6 and Supplementary Note 3, Supplementary Fig. 10), suggesting no obvious issue in the model fitting. In addition, we had initially hoped to consider models, including species-specific slopes between size and $\delta^{13}C$ or $\delta^{18}O$ values as well as offset-intercepts, but current data limitations prevented us from doing so (Supplementary Note 2, Supplementary Fig. 1 and Supplementary Data 1). For example, whilst all the major ocean basins are represented here, there is a paucity of size-specific data for many species and a variable degree of isotopic scatter among them (Table 1). A fruitful avenue for future research would be to design a more balanced sampling regime across species and basins to relax restrictive assumptions of the current approach (e.g., species-specific offsets are expressed via the intercept $a$ and not the size-dependent slope $b$ due to the lack of resolution in the current data, Supplementary Note 2, Supplementary Fig. 1). This would enable a more robust set of predictions to be developed. The current random effect estimates are unable to partition variation due to variables like primary productivity or carbonate ion effects that are not included in our analyses from species-specific offsets. Our model-averaged results are nevertheless rooted in empirical observations and robust statistical methodology[20] (Supplementary Note 2, Supplementary Fig. 6 and Supplementary Note 3, Supplementary Fig. 10), but

should be considered as falsifiable hypotheses that are highly dependent upon the data used.

Ultimately our results provide the first quantification, in any clade, of the impact of shared evolutionary history on $\delta^{13}C$ values. Despite applying the same workflow to $\delta^{18}O$ values, we found little or no impact of shared evolutionary history on that isotopic system. This does not necessarily imply that there are not species-specific vital effects for $\delta^{18}O$ (see paragraph above for caveats), merely that $\delta^{18}O$ vital effects do not vary systematically across the phylogeny as $\delta^{13}C$ does given the current data and models. Benthic foraminifera are also a key substrate for paleoenvironmental reconstructions and it's likely that similar evolutionary effects on $\delta^{13}C$ values will exist in this group as well. However, long-term benthic foraminiferal stable isotope reconstructions are generally derived from a single taxa or genus, as taxa are typically much longer lived than their planktic counterparts and thus, the impact of accumulated evolutionary differences on these records are likely much smaller. Our results also imply that other marine calcifiers such as corals, molluscs and echinoids, which fractionate carbon much more strongly than foraminifera (up to 13‰ vs. 1–3‰) and show strong species-species signals[3, 29–31], will also likely carry a (potentially much larger) phylogenetic imprint meriting investigation. Clearly, evolution matters for organismal geochemistry, and phylogenetic methods provide a promising means to crack open the black box of vital effects.

## Methods

**Stable isotope compilation**. All available size-specific $\delta^{13}C$ and $\delta^{18}O$ data were compiled from the existing literature, with the requirement that all samples have a minimum of three sieve size classes and a total of four data points per species for subsequent model fitting. The available single-specimen data with sufficient size range[32] are included but are averaged within size bins for consistency with the majority of data used here. The final compilation of 3797 data points from the Cenozoic and Cretaceous include the core-top database of modern macroperforate planktonic foraminifera collated by Ezard, et al.[20], multiple paleorecords from published sources[6, 16, 32–56] sources and this study. We have 'good' data coverage between 0–2.4 Ma, ~23 Ma, ~39–42 Ma, and 56–66 Ma.

Taxonomic names were updated, when necessary, according to the following references: Mesozoic Foraminiferal Working Group[57], Paleogene Working Group Atlases[58, 59], and Aze, et al.[15] with modifications to the phylogeny as in Ezard, et al.[20]. We use a morphospecies phylogeny in this study for comparisons between modern and fossil taxa[7] because a phylogeny of the same scale based on genetic data is not available as most species are extinct. In order to assign benthic foraminiferal $\delta^{13}C$ and $\delta^{18}O$ (two environmental predictors), each sample was assigned a planktonic foraminiferal biozone following the Geological Timescale 2012[60].

**Predictors of planktonic foraminiferal size-isotope trends**. Each planktonic foraminiferal species in the data set was assigned a number of environmental, biological and preservational attributes (Supplementary Table 1). Attributes include ocean basin of the core site (Atlantic, Indian, Pacific); primary biome ((sub)tropical vs. transitional/polar); benthic foraminiferal $\delta^{18}O$ and $\delta^{13}C$ values (average within each planktonic foraminiferal biozone); depth habitat (mixed layer, thermocline or sub-thermocline); type of algal photosymbiont (asymbiotic, chrysophyte-bearing or dinoflagellate-bearing); spinosity (spinose vs. non-spinose); wall structure (finely, micro- and macro-perforate); water depth above the core site; and fossil preservation state (modern, excellent, and recrystallized). "Excellent" preservation is equivalent to "glassy" carbonate preservation e.g., ref. [22]. All extinct taxa were assigned as either bearing dinoflagellate symbionts or asymbiotic because of difficulties recognizing chrysophyte-bearing taxa in the fossil record[33]. Background benthic foraminiferal $\delta^{18}O$ and $\delta^{13}C$ values were included to control for long-term changes in the global ocean and climate[1, 19]. We note that while these records capture long-term trends in global climate, in reality they record changes at the high latitudes, where deep water is sourced. Bi-hemisphere deep-water formation is evident since at least the late Eocene, evidence for northern component deep water prior to this is less clear e.g., ref. [61]. Thus, southern-sourced deep waters may have dominated the ocean depths, and thus the environmental signal employed in the older portion of our records. We also tested the effect of replacing water depth above the core site with the paleowater depth by re-running the $\delta^{18}O$ MAM and then comparing the resulting AIC. Inclusion of paleowater depth did not change the amount of variation explained and in fact compromised the model stability so we have retained modern water depths in the main text (Supplementary Note 6). All raw data are in Supplementary Data 1.

We construct non-linear mixed effect models in a stepwise fashion—the initial model backbone is given in Eq. 1, and the explanatory variables can impact each of these parameters. We build up to a full model including different variation patterns for each species (heteroscedasticity). From this full model, we remove explanatory variables sequentially in a reverse stepwise process according to least explanatory power. Supplementary Notes 2 and 3 details each step in this process for $\delta^{18}O$ and $\delta^{13}C$, respectively. This process leaves parsimonious minimum adequate models that use fewer parameters to explain statistically similar amounts of variation in the data set (Supplementary Data 1).

As there is unlikely to be one "true" model, we construct our synthetic ocean core using a model-averaging projection approach. Each model in the simplification sequence has a likelihood and an Akaike Information Criterion score. The latter is associated with an Akaike weight, which sum to 1 for all models being compared. By multiplying each model's predictions for the synthetic ocean core by the model's Akaike weight, and then summing all model predictions, we are able to generate model-averaged estimates for the species-specific offsets. This process is detailed in Supplementary Notes 2 and 3, with graphical visualizations and each species' predicted offset for the synthetic core. The numbers in Supplementary Table 4 are then used to test for the effect of shared evolutionary history on these patterns as detailed in Supplementary Notes 4 and 5.

**Testing for the effect of shared evolutionary history**. The Cenozoic phylogeny of macroperforate foraminifera[15] was used to test the effect of shared evolutionary history on trait expression with two amendments. *Neogloboquadrina incompta* and *Globigerinoides ruber* pink were grafted onto the morphospecies phylogeny, assuming a late Miocene (6 Ma) divergence between *N. incompta* and *N. pachyderma*, and between *G. ruber* pink and white as described in ref. [20]. Since comparable phylogenies are not available for Cenozoic microperforate species or Cretaceous taxa, our test for the effect of shared evolutionary history is limited to 66 of the 105 species in the compilation.

Pagel's $\lambda$[24] only transforms the tips of phylogenies. We therefore used the morphospecies phylogeny with persistent ancestry through speciation (the aM tab in Supplementary Table 4 of ref. [15]) for all analyses to ensure all data is treated equally. Phylogenetic generalized least squares models were fitted using the pgls function in the caper package[62], estimating $\lambda$ by maximum likelihood. Blomberg's $K$[25] was calculated using the phylosig function in the phytools package[63] (Supplementary Note 4).

There are higher-level patterns (over and above the species level) of phylogenetic dependence on the phylogeny, which include isotopically critical aspects such as the presence or absence of photosynthetic algal symbionts (see ref. [64] for their macroevolutionary influence). To explore the impact of these above-the-species-level patterns on $\delta^{13}C$ and $\delta^{18}O$, we re-ran the full model without the biological aspects (symbiosis, spinosity and wall structure). While the reduced model for $\delta^{13}C$ did not converge, Supplementary Note 5 details how our maximum likelihood estimates of $\lambda$ for $\delta^{18}O$ were 0 with (95% confidence intervals between 0 and 0.45) with and without correction for symbiont presence, depth habitat and biome. The similarity between estimates suggests that the biological explanatory variables in our analytical workflow are not biasing our results.

**Code availability**. The full marked up R code and rationale for each step of our current study is available at figshare with DOI 10.6084/m9.figshare.5048854.

**Data availability**. The full raw stable isotope compilation along with ecological, environmental and preservational predictors used in this study is included as Supplementary Data 1.

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

## Acknowledgements

We thank all the authors who made their data freely available. Financial support was provided by a Leverhulme Early Career Fellowship ECF-2013-608 to K.M.E, a NERC Advanced Research Fellowship NE/J018163/1 to T.H.G.E., and a NSF Award #1536604 to P.M.H.

## Author contributions

K.M.E. and P.M.H. assembled the underlying database, and T.H.G.E ran all statistical and phylogenetic tests. All authors contributed to the inception, interpretation and writing of this manuscript.

## Additional information

**Competing interests:** The authors declare no competing financial interests.

