## [Peer Review File · Nature Communications]

Reviewers' comments:

Reviewer #1 (Remarks to the Author):

Evolutionary history biases geochemical inferences of ecology and Environment

Kirsty M. Edgar, Pincelli M. Hull and Thomas H.G. Ezard
[redacted]

Helen Coxall

D13C of planktonic foraminifera is increasingly being put to the test in terms of extracting information about nutrient structure and carbon cycling regimes during past non analogue climate states. This paper claims to have identified an overlooked aspect that explains a large part of the variability in d13C signals in planktic forams (specifically the test size d13C relationship of individual species) namely shared evolutionary history, which the authors conclude must be considered when interpreting the d13C of planktonic foraminifera.

The paper centers around the finding that shared evolutionary history biases geochemical inferences of ecology in planktonic foraminifera. The title is intriguing, although I was immediately wondering exactly what this might mean and how it was going to be implemented. The concept and methodology are sound, the data required to repeat the experiments are provided, and the significance potentially very important and worthy of publication in Nature Geoscience. However, in trying to make the message a general and accessible one, I feel that the detail that would make this a really concrete tangible contribution has been lost or buried within the statistics and summary figures and the core idea thus hard to grasp.

I think this is because from the start this idea of what and, at what taxonomic/phylogenetic level things are shared, and how this translates into practical taxonomy or proxy application goes without clear definition at the start.

If the authors can overcome this then this paper is potentially really interesting and important. The rest of my review expands on what I think is missing in relation to communicating the core concept of 'shared evolutionary histories', and second, about dealing with what I think is the key practical finding, i.e. that this evolutionary/biology 'bias' is most important for d13C, and its application in palaeoceanography.

'Evolutionary relatedness'

I understand the need to carefully account for a range of factors when extracting environmental signals from biological remains but I fail to grasp exactly where the question of evolutionary relatedness of species has been or is a problem or is a new idea.

There are several points in the ms where this concept of relatedness is mentioned but its definition is sufficient for doing what the paper suggests we should be doing, i.e. considering relatedness when inferring palaeoecologies and identifying/using specific taxa to trace water column processes

e.g:

Line 23-25: "Closely related taxa are, on average, more similar than distantly related ones, implying that the changing composition of marine communities through time may systematically bias geochemical interpretations."

Maybe if we were presented with an example of a case where this is a problem it might become clearer. Or how, after the results of this study, we can improve upon ecological inferences.

Also:

Intro paragraph

Lines 21-23

"Of the many factors carefully accounted for when extracting environmental signals from biological remains, the evolutionary relatedness of species is not typically considered.

In what context- do you mean within any one assemblage? From a time period?

Are you saying that assemblages may contain more closely related species at certain times compared to other times? Related at what level of tree hierarchy? How exactly is this connection with the isotopes and the process of making ecological inferences?

Then:

Line 200-: "The variation in $\delta^{13}\text{C}$ offsets largely corresponds with the evolutionary history of taxa. Thus, for long-term analyses of the marine carbon cycle, the phylogenetic signal in $\delta^{13}\text{C}$ vital effects imprints an important signal on the secular trend."

Its still not clear what exactly this phylogenetic signal is, although I understand the principal, and what I should do with it in the future.

This term "for long-term analysis of the marine carbon cycle" I think is key here. If you make this goal more apparent earlier on I think the message might come through better.

As I understand planktonic foraminifera ecology and palaeoecology, there is already an expectation, supported by isotopic data, that fossil species from the same genera have a similar ecology and biology, e.g. Paleocene and Eocene *Morozovella* (~ 11 and 12 species respectively) were mixed layer dwelling species with dinoflagellate symbionts, while co-occurring species of the genera *Subbotina* (5-15 Paleocene and Eocene species respectively) were typically deeper living thermocline dwellers without symbionts. Jumping forward in time, in the modern oceans, we find key generic groups, e.g. *Globorotalia* (~ 12 species) that plankton tow data reveal spend important stages of the life cycle at near thermocline depths while other related species, e.g. *Globigerinoides/Orbulina* (4 species) remain in the mixed layer/photoc zone. We predict that similar ecological niche existed and were occupied by planktonic foraminifera in the past as today but is the point then that the biology of the actual organisms was fundamentally different, i.e. more different than we have been assuming, especially for symbiotic taxa? The paper does go towards this at the end but maybe set this idea up better at the beginning.

There is only one place that we get a sense of what level of relatedness real mention of specific evolutionary relationships is among 3 taxa, (*Morozovella*, *Acarinina* and *Igorina*, and of these the illustrative figure helping communicate this idea is in the Supplementary Information. (SI Figure 12). I suggest bringing this figure out of the SI and combining it in some way with the existing main text figure version so we get a sense of revolutionary groups in the main text.

Line 47 'biology' – too vague. Do you mean biology of the long extinct fossils? Biology of the whole ocean (marine organic pump vs solubility pump, or peculiarities of individual taxa?

Line 50: Somewhere in the paper you should remind us that that planktonic forams calcify surprisingly close to ambient seawater chemistry their d^{13}C compared to other calcifying groups such as corals and molluscs.

Line 56: "If closely related species have more similar vital effects than distantly related species" ...again at what taxonomic level

'The time related shifts in high latitude d^{13}C '

High latitude (i.e. benthic values?) d^{18}O and d^{13}C variables are used as overall

environment/climate indicators in the correlation experiments. I'm interested to know more about what values were used to represent this over time, especially for d13C, which turns out not to have a strong explanatory effect on planktic foram shell size d13C relationships. This is particularly relevant to d13C in the Neogene, where we see the largest ranges of planktic foram size related d13C.

Importantly, the Cramer et al. 2009 stable benthic isotope compilation shows that high latitude DIC d13C became on average lower for the whole ocean from 5 Ma, while the difference between Atlantic and Pacific deep ocean (high latitude surface ocean) end members increased markedly.

This is telling us something significant about changes in the carbon system at this time. Is telling us something about ocean biology generally? And are you missing some ocean wide biology effect related to this by using a single 'representative' d13C value in the correlation whereas in reality the oceans become highly heterogeneous?

- differences in air-sea gas exchange in a cooling ocean?
- consequence of enhanced biological and solubility pumps effecting the marine biopump, C3/C4 plants transition?
- or terrestrial biosphere controls the isotopic composition of atmospheric CO2

Comments on the figures

Figure 1

This is a nice figure. Huge difference for time periods very clear. Cool. This is a really new perspective.

Specific info on the position of different 'related' groups is missing. Can you have a version in the SI showing example clades? Eg. Some ancient ones and some younger ones –Neogene /Quat.

Figure 2: Which of the explanatory variables represented in this figure correspond to the concept of 'evolutionary relatedness'? Can you annotate in some way or indicate clearly in the text/caption.

For the lower panel, d13C, the preservation variable is large but 'transparent' – I don't quite understand the significance of the 'does not fit' qualifier. Is preservation an important explainer of the d13C variability or not? Please clarify.

Regarding the relationship between size specific d13C and high latitude d13C. As mentioned in the text, there are

Figure 3

Does the preservation aspect not dominate the Paleocene Eocene data set? Or is it just for d18O? Are there ooze and glass forams assemblage data mixed in together?

Can you explain what those coloured 'eyebrows' are – are they synthetic isotope size trajectories? Bit more information needed.

The Fig 3 caption says that line thickness is proportional to the available size data for each species –but isn't this the model output? What is controlling the differences between eyebrows orientation/tail thickness within and between time slices Eocene and Cretaceous compared (i.e. Eocene and Cretaceous mainly skinny at the tail getting fatter with decreasing d18O).

What's the single giant eyebrow in the Neogene?

Lines 140-146: Maybe for carbon but not so for oxygen. Explain. Is this a consequence of combining 'glass and non glassy data sets. (fundamentally different).

137-138

I don't understand the concept of your synthetic core and why it has the three time periods. Do you mean its theoretically sampling the three time periods (imaginary long core cutting deep and through multiple layers) at a single oceanic setting? Does it build in a 'preservation effect' or are they forams all glassy/ ooze?

Line 142: Igorina is not shown in sup info fig 12 so not really accurate to include it in the group example listed here.

Line 157: what's a star phylogeny –clarify. Star-shaped, do you mean any shape, why a star? Do you mean as a contrast to a nested tree?

SUPPLEMENTARY INFO

P. 34 Figure SI12. This is the only figure where we get to see the 'relatedness' factor in any kind of graphical form. What was the rationale for picking these genera as examples? Are these the ones with data for several species ?

Incidentally, I was repeatedly going back to Fig 12 SI to get something tangible with regards taxa and their position in isotope-size trend space', suggesting it should be in the main pain in some form.

P. 38 sup info: this line in the sup info is crucial to the whole paper and should be buried in here. "To assess the evidence if species differences are more involved than can be explained by random intercepts alone, we need a phylogeny that contains a hypothesis of the inferred evolutionary history of the clade. Macroperforate planktonic foraminifera are one of the few groups with a near-complete phylogeny (Aze et al., 2011)."

Reviewer #2 (Remarks to the Author):

This a finely written paper that compiled many stable isotope data from multiple sources across a long time interval, but I am not convinced that it includes significant insights that merit publication in Nature Communications. The statement on line 62 "Of the many factors carefully accounted for when extracting environmental signals from biological remains, the evolutionary relatedness of species is not typically considered" is puzzling as most workers usually target shallow vs. deep mixed layer and thermocline dwelling planktonic foraminifera by selecting species that are within genus groupings for their time series studies. Such determinations are inferred from within sample comparisons of interspecies offsets for samples across time series, but the depth ecology assignments are often uncertain for extinct species in the absence of independent measurements of the vertical temperature and carbon isotopic profile. Species that are assigned as the "shallowest dwelling" may yield the most negative $\delta^{18}\text{O}$ and most positive $\delta^{13}\text{C}$ relative co-occurring species measured, but we can't know whether it actually did live in the upper mixed layer. The same applies to extinct species that get assigned as thermocline dwellers. Measurements of a higher diversity of species within samples and increased sampling resolution may improve confidence in depth ecology assignments for different taxa, but studies have also revealed that some species show a stratigraphic shift in their depth habitat with stable isotope signatures that are significantly offset from other species in the same genus or lineage. So the story can get quite complex.

I am also concerned that each of the three grouped time intervals (Late Cretaceous, Paleogene, early Cenozoic, late Cenozoic) contains profound variations in ocean temperature and vertical structure of the water column, but there is no allowance for such complexity in the data analyses, despite the fact that such environmental changes would have strongly influenced interspecies offsets in stable isotopic values. In my opinion the dataset is too incomplete and assumptions made from it are too simplistic to discern the relative importance of environment, life strategy,

phylogeny and preservation in influencing the $\delta^{13}\text{C}$ and $\delta^{18}\text{O}$ values of foraminifera across the past 120 million years.

Additional concerns:

- The text states that the study encompasses data spanning the past 120 m.y. but the oldest data in data table is 106.98 m.y.
- The Cretaceous data set has several time gaps of 3-5 m.y., resulting in incomplete characterization of lineages that span the missing time intervals
- A number of Cretaceous taxa have never been tested for size-related changes in $\delta^{13}\text{C}$ so the assumption that they are asymbiotic may be incorrect
- An explanation for the data table organization, header abbreviations, numerical depth ecology assignments, preservation rating assignments is missing
- It is not clear why present water depth is listed when paleowater depth would be more useful
- The sources of data used in the analyses is very heterogeneous; there is no accounting for paleolatitude and local/regional paleoceanographic conditions, which would have influenced relative stable offsets between species (e.g., high clay runoff in nearshore settings would have caused high light attenuation and shallowing of depth habitats; depth habitats would have been influenced by paleolatitude).
- Some incorrect pore size assignments (e.g., *Biticinella breggiensis* is macroperforate but is listed as finely perforate; *Marginotruncana sinuosa* listed as microperforate but is macroperforate)
- Incorrect genus assignment of *Hedbergella simplex* (should be *Clavihedbergella*)
- There is no reference list for sources of data in data table

Reviewer #3 (Remarks to the Author):

Edgar et al. test how evolutionary changes may have impacted carbon and oxygen isotope ratios in planktonic foraminifera over the last 120 years. They find that species-specific effects have a large influence on average isotope compositions and on isotopic variability. The authors use statistical methods to assess what predictive factors best describe the histories of $\delta^{13}\text{C}$ and $\delta^{18}\text{O}$ given by planktic foraminifera. The impact of species specific effects is more clearly seen for $\delta^{13}\text{C}$, in line with what is already understood about the importance of vital effects for that isotope system. However, the statistical tests performed in this study more quantitatively demonstrate the importance of considering vital effects in interpreting paleo- $\delta^{13}\text{C}$ from planktic foraminifera. While the importance of foraminiferal growth environment, preservation, size, and vital effects are generally known, this study is significant in that it attempts to tease apart the relative impacts of each. Overall, I think this paper is clearly written and requires minimal revisions. This paper will be important for paleoceanographers and paleoclimatologists who seek to interpret geochemical results derived from planktic forams. In addition, I think this study can serve as a template for investigating other geochemical parameters (Mg/Ca, Sr/Ca, etc.) that are impacted both by environment and biology. The major questions I have are related to the assumption that species-specific effects are only applied to the intercept ('a') and the residuals in equation 1.

Introductory paragraph: I found that some of the conclusions stated in the introductory paragraph were a bit inconsistent with the results and conclusions stated later in the paper. In particular, the introduction doesn't distinguish between results for $\delta^{18}\text{O}$ and $\delta^{13}\text{C}$ (i.e. line 29-30: "Of the four, the most important is species-specific effects on both the isotopic means and variability"), whereas later in the paper the results clearly imply that species-specific effects are more important for $\delta^{13}\text{C}$ than $\delta^{18}\text{O}$. It would be helpful if the text were clarified in the introductory paragraph.

Lines 108-110: The authors state that species-specific effects are only applied to the offset term ('a') and residuals (epsilon) in equation 1. While the authors give general reasons for doing this

(lack of sufficient data), some of the results and interpretation seem like they could be strongly influenced by this assumption, giving a circularity (at least as it seemed to me) to some of the major take-homes of the paper. While every model has assumptions underpinning it, I do think it would be useful to more clearly state the implications of this assumption in the main text, and perhaps include some statement regarding how the results might change if vital effects were applied to the other terms as well.

Line 122: Perhaps AIC could also be defined in the main text?

Line 132: I think this conclusion is an interesting one, and if there's sufficient room, it would be nice to explain a little bit further since I didn't find the sentence to be particularly clear (although there is a bit more explanation later in the text). Is the key point here that the expression of species-specific $\delta^{13}\text{C}$ effects depends on how climatic and preservational factors influence the slopes and absolute values? I am also not sure, again, whether this conclusion is a result of the assumption imposed that vital effects are only applied to the offset term 'a'.

Lines 231 and the last paragraph of the paper: It would be useful if the authors included another statement about timescale and if they were more explicit about the impacts of turnover for evaluating carbon cycle changes. First, the data are broken up into relatively large chunks of geologic time (10s of millions of years), and the analysis generally implies that the conclusions are relevant over evolutionary timescales. But I wonder how the data would group if you looked at data in finer intervals? Second, there is a statement cautioning that evolutionary changes could obscure environmental signatures at times when turnover is high (mass extinctions, etc.), but it seems like the authors may be able to provide a more concrete estimate for how important this would be during those intervals (i.e. could the variations observed in records be entirely due to evolutionary shifts)?

Reviewer #4 (Remarks to the Author):

This manuscript explores the potential for evolutionary history to affect foram isotopic records over the last 120 million years. The authors find a strong phylogenetic signal for carbon isotope vital effects, but none for oxygen isotopes. This is at least partially reassuring for those who use these isotopes. The manuscript is well written.

The ideal test for vital effects would compare the isotopic composition of the foram tests with the water they grew in. This is obviously only available for extant species (and may be difficult even for these). For the fossil forams, data on the isotopic composition of the water are not available, and instead the authors have to rely upon the biogeographic affiliation and a benthic foram oxygen isotope derived estimate of global climate. This probably limits the potential to find a phylogenetic signal in the oxygen isotopes.

In their model, the authors include foram characteristics such as wall morphology as predictors. As these characteristics are conserved, these predictors probably explain some of the variance that could also be explained by phylogenetics. This could also explain why there is no phylogenetic signal in the oxygen isotopes. Whether it is useful to include these predictors depends on the question being asked. If the authors are interested in the full phylogenetic signal, they should probably be excluded. If they are interested in the residual phylogenetic signal after obvious biological predictors are accounted for, they should be included. For applied work, the second is probably more useful. This could be explored.

It is not immediately obvious why the authors are using a non-linear mixed effects model when there are no nonlinear terms in their model - all the terms, including the quadratic - are additive.

This could be explained.

It would be useful to discuss the magnitude of the vital effect revealed by the analysis (whether there is a phylogenetic signal or not), and how large it is relative to the apparent signal in isotopic records.

I don't really understand figure 3. I think it is the phrase "available size data" which is confusing me as all the data included had to have size data.

I greatly appreciate the authors providing the Rmarkdown output with all the scripts. I would recommend, however, that this code and the data are archived, for example in a github repository, rather than making it available from the authors by request.

Response to Reviewer's Comments for manuscript "Evolutionary history biases inferences of ecology and environment from $\delta^{13}\text{C}$ but not $\delta^{18}\text{O}$ values"

For ease, we have included the original comments from each of the reviewers that required an action in bold and our response to comments beneath in blue.

Reviewer #1 (Helen Coxall):

- **D13C of planktonic foraminifera is increasingly being put to the test in terms of extracting information about nutrient structure and carbon cycling regimes during past non analogue climate states. This paper claims to have identified an overlooked aspect that explains a large part of the variability in d13C signals in planktic forams (specifically the test size d13C relationship of individual species) namely shared evolutionary history, which the authors conclude must be considered when interpreting the d13C of planktonic foraminifera. The paper centers around the finding that shared evolutionary history biases geochemical inferences of ecology in planktonic foraminifera. The title is intriguing, although I was immediately wondering exactly what this might mean and how it was going to be implemented. The concept and methodology are sound, the data required to repeat the experiments are provided, and the significance potentially very important and worthy of publication in Nature Geoscience.**

Thank you!

- **However, in trying to make the message a general and accessible one, I feel that the detail that would make this a really concrete tangible contribution has been lost or buried within the statistics and summary figures and the core idea thus hard to grasp. I think this is because from the start this idea of what and, at what taxonomic/phylogenetic level things are shared, and how this translates into practical taxonomy or proxy application goes without clear definition at the start. If the authors can overcome this then this paper is potentially really interesting and important.**

Thank you for these focused suggestions for improvement - these points really drove home the gap in our current explanatory approach and we have taken the following steps to correct the problem:

- Added a clear definition of phylogenetic relatedness to the introduction;
- Introduced two new figures – one of how $\delta^{13}\text{C}$ and $\delta^{18}\text{O}$ offsets differ from *Subbotina velascoensis* as a representative case study (see right new Fig. 5), and a general summary figure for all differences from all species (new Fig. 6);
- These new figures are supported by further statistical analyses, which generalize the differences in the case study figure (right). We show how $\delta^{13}\text{C}$ offset differences among species are larger than for $\delta^{18}\text{O}$, and furthermore that $\delta^{13}\text{C}$ offsets correlate positively with increasing evolutionary distance, whereas $\delta^{18}\text{O}$ do not. In other words, these new analyses suggest that the patterns we display in the case study hold across the phylogeny;
- Provide an explicit example of what our approach adds 'on top of' the methods that are already used (e.g., vital effects, habitat effects, seawater

chemistry/condition effects) and included suggestions for how to apply this approach moving forward.

- **The rest of my review expands open what I think is missing in relation to communicating the core concept of ‘shared evolutionary histories’, and second, about dealing with what I think is the key practical finding, i.e. that this evolutionary/biology ‘bias’ is most important for $\delta^{13}\text{C}$, and its application in palaeoceanography.**

We take each point in turn.

- **‘Evolutionary relatedness’ - I understand the need to carefully account for a range of factors when extracting environmental signals from biological remains but I fail to grasp exactly where the question of evolutionary relatedness of species has been or is a problem or is a new idea.**

At present, paleoceanographers and geochemists acknowledge that vital effects are important and state they account for them through symbiont effects, depth habitat, and measuring isotopic offsets between specific species. Closely related species within the same genus are chosen for long-term environmental reconstructions and then across extinction boundaries or between low-and-high latitude assemblages we jump to other clades on the basis of symbiont ecology, depth habitat, and relative isotopic differences. Geochemists therefore use an implicit evolutionary hypothesis, but do not test it formally. We seek to move beyond this ad-hoc incorporation of vital effect dependence to an explicitly evolutionary setting (we now make this point on lines 57-91), embedding changes among species and correlating biological effects in mathematical models of evolutionary divergence. Our revision also now makes explicit how this formalization helps with practical application (lines 64-73; 373-376) and what additional data is required to probe hypotheses of the chemical impacts of evolutionary relatedness more comprehensively (lines 378-402).

So why is this a new idea? So why does it matter that geochemists have two different implicit assumptions about the influence of evolutionary history on geochemical signals but leave them untested?

These two questions are at the heart of the matter that we were trying to communicate. In our analyses, we show that there is additional information (i.e., additional systematic variation in $\delta^{13}\text{C}$ values) that can be explained by an evolutionary hypothesis based on species’ shared evolutionary history. This means that the similar species chosen for geochemical reconstructions violate the statistical assumption of independence among data points. We would like to know how non-independent different species are, and use the Aze et al. (2011) planktic foraminiferal phylogeny as a backbone for these calculations. We argue (see revised sections on lines 55-93) that using explicit evolutionary hypotheses in geochemistry matters because (1) it provides information about systematic biases (i.e., unaccounted for variance) in datasets that span large assemblage turnovers (in space or in time); and (2) it provides a tool for directly testing the influence of evolution as is standard in all other disciplines using similar comparative analyses.

- **There are several points in the ms where this concept of relatedness is mentioned but its definition is sufficient for doing what the paper suggests we should be doing, i.e. considering relatedness when inferring palaeoecologies and identifying/using specific taxa to trace water column processes e.g: Line 23-25: “Closely related taxa are, on average, more similar than distantly related ones, implying that the changing composition of marine communities through time may systematically bias geochemical interpretations.” Maybe if we were presented with an example of a case where this is a problem it might become clearer. Or how, after the results of this study, we can improve upon ecological inferences.**

Excellent point. We have already re-worked our introduction based on the reviewers’ earlier comments such that the concept of relatedness and applications of this work are more apparent (lines 57-91).

- **Lines 21-23 - “Of the many factors carefully accounted for when extracting environmental signals from biological remains, the evolutionary relatedness of species is not typically considered. In what context- do you mean within any one assemblage? From a time period? Are you saying that assemblages may contain more closely related species at certain times compared to other times? Related at what level of tree hierarchy?**

We have subsequently modified the main text to address the reviewers earlier comments and now explicitly state in lines 76-85-“Closely related foraminifera often share multiple morphological and ecological

characteristics¹⁵. For example, extant shallow dwelling, dinoflagellate-bearing planktonic foraminiferal species all occur in one branch of the phylogeny¹⁵. Thus, one might expect two species from the same genus to have more similar vital effects than two species from different genera. For instance, shallow water or photosymbiotic species might have additional metabolic adaptations to their environment, offsetting their isotopic composition from environmental values. However, it's important to note that species from neighboring genera may be more similar to each other if they are more closely related (in terms of time or number of speciation events since last common ancestor) than two more distantly related species within the same genus."

- **How exactly is this connection with the isotopes and the process of making ecological inferences?**

This is a hugely important question that we previously side-stepped for reasons of brevity. Reviewer Four's question has made us directly address the question in the manuscript, so we will provide a short explanation here and we refer you to the new text (lines 284-291) and our later response to that reviewer. We show that after all the known biological and ecological traits are accounted for, there is STILL a phylogenetic signal in the $\delta^{13}\text{C}$ data and not for the $\delta^{18}\text{O}$ data. The ecological predictors are weaker predictors than the shared evolutionary response in our current data (see section 3.6 in the Supplementary Materials and later in this response for analyses), but fully testing this question really does require much more and much more systematic data across the phylogeny. We hope that our current work catalyzes this effort to collate new and comprehensive isotope data to crack open the black box of vital effects from an evolutionarily informed perspective.

- **Line 200-: "The variation in $\delta^{13}\text{C}$ offsets largely corresponds with the evolutionary history of taxa. Thus, for long-term analyses of the marine carbon cycle, the phylogenetic signal in $\delta^{13}\text{C}$ vital effects imprints an important signal on the secular trend." Its still not clear what exactly this phylogenetic signal is, although I understand the principal, and what I should do with it in the future. This term "for long-term analysis of the marine carbon cycle" I think is key here. If you make this goal more apparent earlier on I think the message might come through better.**

The phylogenetic signal in our study refers to the species-specific offsets (formally, the random effects on the intercept term (a) in equation 1-line 147) from the global trends experienced by the overall clade as it responds to a changing environment. This is the first time that these have been identified and quantified. The longer-term vision for this work is to get to a point where vital effects can be predicted accurately for any species, any size fraction, any time interval and thus, be used to 'correct' isotope records and extract a primary environmental signal (now stated explicitly on lines 378-380). Our study represents an initial step towards this ambitious goal by first showing that it is both needed and useful. We now include a more step-by-step approach to the analyses and have introduced three new figures (4-6) to better link results to discussion.

In the process of these analyses, we discovered new insights into the controls on isotope records, and that there is currently insufficient data available to generate high confidence species-specific offsets to 'correct' isotope records – in particular, we would like to expand our treatment of species-specific offsets to include linear slopes as well as y-intercepts, but the current data are not sufficient if we also seek to correct for well-understood environmental and biological influencers of isotopic composition (lines 378-402).

- **As I understand planktonic foraminifera ecology and palaeoecology, there is already an expectation, supported by isotopic data, that fossil species from the same genera have a similar ecology and biology, e.g. Paleocene and Eocene *Morozovella* (~ 11 and 12 species respectively) were mixed layer dwelling species with dinoflagellate symbionts, while co-occurring species of the genera *Subbotina* (5-15 Paleocene and Eocene species respectively) were typically deeper living thermocline dwellers without symbionts. Jumping forward in time, in the modern oceans, we find key generic groups, e.g. *Globorotalia* (~ 12 species) that plankton tow data reveal spend important stages of the life cycle at near thermocline depths while other related species, e.g. *Globigerinoides* /*Orbulina* (4 species) remain in the mixed layer/photoc zone. We predict that similar ecological niche existed and were occupied by planktonic foraminifera in the past as today but is the point then that the biology of the actually organisms was fundamentally different, i.e. more different than we have been assuming, especially for symbiotic taxa? The paper does go towards this at the end but maybe set this idea up better at the beginning.**

Good point. In brief, we are implying that the 'biology' of foraminifera even with inferred similar ecologies

likely changed through time and in this we are supposing this could be the metabolism: i.e., basal metabolic rate; basal photosynthetic rate; slope of temp. sensitivity of both (now in line 376). But it could also be a whole suite of other factors. For instance, many organisms have the ability to suppress their metabolisms during times of food scarcity or environmental stress. In the open ocean, many taxa spend a large amount of their 'life' essentially in this 'suppressed' or 'off mode'. To what extent foraminifera do this is unclear (e.g., Ross and Hallock 2016. *J. Foram. Research*) although it is known they have the capacity to do so. How this affects what exactly is recorded by their geochemistry is an interesting question. Regardless, you are right to say that there are important aspects of biology, like basal metabolic rate, that are currently assumed to be identical across all species (and times) despite many good reasons to expect this to not be the case. So we've amended the front end of the paper accordingly (lines 61-73).

- **There is only one place that we get a sense of what level of relatedness real mention of specific evolutionary relationships is among 3 taxa, (Morozovella, Acarinina and Igorina, and of these the illustrative figure helping communicate this idea is in the Supplementary Information. (SI Figure 12). I suggest bringing this figure out of the SI and combining it in some way with the existing main text figure version so we get a sense of revolutionary groups in the main text.**

Excellent suggestion. We have now made a selection of key figures (including that mentioned by the reviewer) showing the groupings as additional panels in Figure 4. We have also created a new figure (Figure 5) that explicitly shows the relationship between evolutionary relatedness (as both time and number of speciation events to the last common ancestor) and species-specific offsets for a specific case study *Subbotina velascoensis* (see opening page of this response). We demonstrate the generality of this case study result in new Figure 6 as well as via statistical tests.

- **Line 47 'biology' – too vague. Do you mean biology of the long extinct fossils? Biology of the whole ocean (marine organic pump vs solubility pump, or peculiarities of individual taxa?)**

We're referring to organismal biology, e.g., respiration, photosymbiosis, calcification etc. here and have amended the text (lines 41-43) to make this clear.

- **Line 50: Somewhere in the paper you should remind us that that planktonic forams calcify surprisingly close to ambient seawater chemistry their d13C compared to other calcifying groups such as corals and molluscs.**

This is a good point. Marine calcifiers (e.g., corals, molluscs and echinoids) are characterized by much stronger biological carbon isotope fractionation than foraminifera (up to 13 ‰ versus ~ 1-3 ‰) but also species-specific values (e.g., Weber and Raup, 1966; Stanley and Swart, 1995; McConnaughey, 1989; Birch et al., 2012). Thus, there is potential for very large and variable offsets through time as a function of phylogeny. Our findings may thus, suggest a similar evolutionary control on $\delta^{13}\text{C}$ values in other marine calcifying groups. This information and the wider implications are now included on lines 409-413.

- **Line 56: "If closely related species have more similar vital effects than distantly related species" ...again at what taxonomic level**

At all taxonomic levels – see lines 78-88.

- **'The time related shifts in high latitude d13C' - High latitude (i.e. benthic values?) d18O and d13C variables are used as overall environment/climate indicators in the correlation experiments. I'm interested to know more about what values were used to represent this over time, especially for d13C, which turns out not to have a strong explanatory effect on planktic foram shell size d13C relationships. This is particularly relevant to d13C in the Neogene, where we see the largest ranges of planktic foram size related d13C. Importantly, the Cramer et al. 2009 stable benthic isotope compilation shows that high latitude DIC d13C became on average lower for the whole ocean from 5 Ma, while the difference between Atlantic and Pacific deep ocean (high latitude surface ocean) end members increased markedly. This is telling us something significant about changes in the carbon system at this time. Is telling us something about ocean biology generally? And are you missing some ocean wide biology effect related to this by using a single 'representative' d13C value in the correlation whereas in reality the oceans become highly heterogeneous? -differences in air-sea gas exchange in a cooling ocean? -consequence of enhanced**

biological and solubility pumps effecting the marine biopump, C3/C4 plants transition?- -or terrestrial biosphere controls the isotopic composition of atmospheric CO2

[1] We used the average benthic foraminiferal $\delta^{18}\text{O}/\delta^{13}\text{C}$ record from the compilation of Cramer *et al.* (2009) and Friedrich *et al.* (2012) within the planktic foraminiferal biozone in which each data point fell. This was previously described in the Methods but now appears in the main text for clarity (lines 142-144). [2] The reviewer raises an important issue regarding the evolution of ocean chemistry. For our part, we currently account for any spatial variability in ocean $\delta^{13}\text{C}$ by including basin as one of our predictor variables – this coarse proxy is of relatively minor importance (Fig. 2) and is another area where systematic data collection efforts in the future would be truly valuable (we make this point explicitly on lines 390-399).

- **Figure 1 - This is a nice figure. Huge difference for time periods very clear. Cool. This is a really new perspective.**

Thanks!

- **Specific info on the position of different ‘related’ groups is missing. Can you have a version in the SI showing example clades? Eg. Some ancient ones and some younger ones –Negene /Quat.**

Agreed - there are some distinctive patterns within the dataset that provide strong visual support for our story and thus, we have taken steps to ensure that these are more apparent in the main text (e.g., new Fig. 4, the *Subbotina velascoensis* example on page 1 of this response = Fig. 5 in the revision). We also have Figures S12 and S13 in the supplement that further expand on these patterns.

- **Figure 2: Which of the explanatory variables represented in this figure correspond to the concept of ‘evolutionary relatedness’? Can you annotate in some way or indicate clearly in the text/caption.**

The goal of Figure 2 is to show the relative contributions of the different ‘known’ variables to explaining variation in isotopic signatures – it does not show evolutionary relatedness. Evolutionary relatedness is assessed while controlling these other influencers of $\delta^{13}\text{C}$ and $\delta^{18}\text{O}$ values. This should now be much clearer from our step-by-step methodology in e.g., lines 140-109 and 113-203.

- **For the lower panel, d13C, the preservation variable is large but ‘transparent’ – I don’t quite understand the significance of the ‘does not fit’ qualifier. Is preservation an important explainer of the d13C variability or not? Please clarify.**

One interpretation of the ‘Does not fit’ is that preservation is an important predictor of $\delta^{13}\text{C}$ -size relationships. For clarity we have now replaced ‘does not fit’ with ‘critical to model’ as our findings are consistent with empirical work suggesting that with increasing post-mortem alteration the $\delta^{13}\text{C}$ signal within and between species is homogenized (e.g., Pearson *et al.*, 2001; Edgar *et al.*, 2015).

- **Regarding the relationship between size specific d13C and high latitude d13C. As mentioned in the text, there are**

The reviewers comment is not clear here and so no action has been taken at this stage.

- **Figure 3 - Does the preservation aspect not dominate the Paleocene Eocene data set? Or is it just for d18O? Are there ooze and glass forams assemblage data mixed in together?**

No. Our dataset contains a mix of recrystallized (altered) and very well preserved (glassy) material in both the Cretaceous and throughout the Paleogene – the high relative importance of preservation on these relationships is shown in Figure 2. Preservation was classified as a three-way categorical variable (Glassy, Altered and Modern) because the far greater size of many modern foraminifera skewed results. Because we included preservation as a variable in our model, our resulting model-averaged predictions (Fig. 3) account for the influence of preservation insofar as they are represented by our data compilation. Within each time interval, the range of oxygen isotope offsets is comparable to that of the carbon isotope offsets (see Table S4, Fig. S13) and the presence of such variation in both datasets reassures us that preservation is not preventing us from detecting phylogenetic structure in the oxygen offsets. We make this point on lines 314-317 in the revised version, albeit for the more generalized case of all explanatory variables and not just preservation.

- **Can you explain what those coloured ‘eyebrows’ are – are they synthetic isotope size trajectories? Bit more information needed.**

The main text (lines 191-203) and figure caption have been amended to make this clearer

- **The Fig 3 caption says that line thickness is proportional to the available size data for each species –but isn't this the model out put?**

We make model-averaged predictions across the range of body sizes for which input data is available to avoid extrapolation. The line thickness is intended to show the size-dependence for each species to identify the start and end of each trajectory through isotopic space, which we have added to the legend for clarity.

- **What is controlling the differences between eyebrows orientation/tail thickness within and between time slices Eocene and Cretaceous compared (i.e. Eocene and Cretaceous mainly skinny at the tail getting fatter with decreasing $\delta^{18}\text{O}$).**

The orientation in the eyebrows follows the progression of predicted species-specific isotopes from the smallest body size with data for that species (narrow end) to the largest body size with data for that species (fat end) – the modified figure caption mentioned in the preceding response helps clarify this point in our revision.

- **What's the single giant eyebrow in the Neogene?**

Menardella menardii – there is a much larger range of size-specific data available for this species than all others, which means it is extrapolated beyond all other ranges. The panel below plots the raw $\delta^{18}\text{O}$ against $\delta^{13}\text{C}$ data with point size proportional to mean sieve size):

Figures S6 and S10 show how the statistical approach taken to correct for the biases in this plot (e.g, different environmental locations, different studies, etc.) yields acceptable diagnostic plots.

The curvature, and length of the eyebrow, in this species are excessive, however, because they extrapolate far beyond the size range for all other species (*M. menardii* extends to 1350 μm , with the second largest *O. universa* and *G. sacculifer* at 975 μm). This extrapolation is suboptimal, so in the revision we have followed the approach taken in our Palaeoceanography paper Ezard *et al.* (2015) and restrict the projected range for *M. menardii*. In the revised figures, we plot up to 1150 μm and state this point in the revised supplementary information when providing code that generates this figure.

- **Lines 140-146: Maybe for carbon but not so for oxygen. Explain. Is this a consequence of combining 'glass and non glassy data sets. (fundamentally different).**

As mentioned above the predictions shown in Figure 3 are after controlling for 10 predictor variables (Figure 2), which includes preservation state. The absence of a strong temporal offset in oxygen is unlikely to be an artifact of mixing preservation states given the inclusion of a preservation variable and the large offsets still observed between species.

137-138 - I don't understand the concept of your synthetic core and why it has the three time periods. Do you mean its theoretically sampling the three time periods (imaginary long core cutting deep and through multiple layers) at a single oceanic setting? Does it build in a 'preservation effect' or are they forams all glassy/ ooze?

Apologies, we now more clearly explain in the text (lines 191-203) the synthetic curves purpose and how they were generated. In essence, the generalized linear modeling approach fits statistical models to the data using various information on ecology, environment and preservation (Figure 2). These explanatory variables allow us to tweak the input settings and generate predictions for all species in a "common currency" that facilitates comparison. In this way, species with different depth habitats still have different depth habitats, but the model-averaged predictions imagine all exist in a core with Very Good preservation in the North Atlantic. This is the synthetic (made-up) part. We can then directly compare all of the data from different time slices and environments in the same plot. The three time periods are colored differently for visualization only (see revised figure legend), but were analyzed together. The current data cannot be used to infer time periods because of the strong covariation between time and wall structure – all of the finely perforate forms are in the Cretaceous, while the overwhelming majority of macroperforates are in the Cenozoic.

Line 142: Igorina is not shown in sup info fig 12 so not really accurate to include it in the group example listed here.

It is now included in new Figure 4, which highlights the Truncorotalidae's isotopic isolation.

Line 157: what's a star phylogeny –clarify. Star-shaped, do you mean any shape, why a star? Do you mean as a contrast to a nested tree?

We have now added text to the main manuscript to explain this idea directly (line 277) but the short answer is 'yes', we do mean a star phylogeny in contrast to a tree. A star phylogeny is effectively one evolutionary hypothesis that posits that the relationships between taxa carry no additional information regarding the trait in question. It is effectively the model of 'no evolutionary signal' – any species is equally likely to be equally related (and therefore, here, isotopically similar) to any other species.

SUPPLEMENTARY INFO

- **P. 34 Figure SI12. This is the only figure where we get to see the 'relatedness' factor in any kind of graphical form. What was the rational for picking these genera as examples? Are these the ones with data for several species? Incidentally, I was repeatedly going back to Fig 12 SI to get something tangible with regards taxa and their position in isotope-size trend space', suggesting it should be in the main pain in some form.**

Good point – to address this issue (also raised earlier) we have introduced fig 5 (page 1 of response), which explicitly shows the clear relationship between offsets and evolutionary relatedness. We have also moved several key panels for key Neogene and Paleogene taxa from Fig S12 into the main text (Fig. 4) and added an additional cross plot to the supplement (Fig. S13) of just the species-specific random effects for genera with size-specific data available for more than three species are shown here to highlight clumping of isotope values.

- **P. 38 sup info: this line in the sup info is crucial to the whole paper and should be buried in here. "To assess the evidence if species differences are more involved than can be explained by random intercepts alone, we need a phylogeny that contains a hypothesis of the inferred evolutionary history of the clade. Macroperforate planktonic foraminifera are one of the few groups with a near-complete phylogeny (Aze et al., 2011)."**

Agreed – see line 55 in the revised text.

Reviewer #2 (Remarks to the Author):

- **This is a finely written paper that compiled many stable isotope data from multiple sources across a long time interval, but I am not convinced that it includes significant insights that merit publication in Nature Communications. The statement on line 62 “Of the many factors carefully accounted for when extracting environmental signals from biological remains, the evolutionary relatedness of species is not typically considered” is puzzling as most workers usually target shallow vs. deep mixed layer and thermocline dwelling planktonic foraminifera by selecting species that are within genus groupings for their time series studies.**

This is a similar point to that raised by reviewer 1 – see above for detailed revisions that we do not repeat in full for brevity. Note in particular the new paragraph italicized on page 2 of this response and lines 57-73 in the revision that agree that while the concept of evolutionary relatedness is implicitly used by paleoceanographers, it is done on an ad hoc basis that does not fully embrace the power of modern comparative approaches for phylogenetic analysis (stated explicitly on line 72) underpinned by an explicit evolutionary framework (Cooper *et al.* 2016). In this paper, we introduce those approaches to isotope geochemistry and how they can help us make more precise and more accurate predictions of vital effect offsets in the future (motivation lines 57-73).

- **Such determinations are inferred from within sample comparisons of interspecies offsets for samples across time series, but the depth ecology assignments are often uncertain for extinct species in the absence of independent measurements of the vertical temperature and carbon isotopic profile. Species that are assigned as the “shallowest dwelling” may yield the most negative $\delta^{18}\text{O}$ and most positive $\delta^{13}\text{C}$ relative co-occurring species measured, but we can’t know whether it actually did live in the upper mixed layer. The same applies to extinct species that get assigned as thermocline dwellers. Measurements of a higher diversity of species within samples and increased sampling resolution may improve confidence in depth ecology assignments for different taxa, but studies have also revealed that some species show a stratigraphic shift in their depth habitat with stable isotope signatures that are significantly offset from other species in the same genus or lineage. So the story can get quite complex.**

While we happily agree that reality is complex (e.g., Figure 2), this particular example (depth specific changes in ontogeny) would not affect our hunt for a phylogenetic signal given that we are looking for this factor in the intercept term (line 158 in our revised manuscript). With regard to the question of ontogenetic changes in depth habitat, indeed some morphospecies do indeed show a change in their depth habitat through time, like *Denotoglobigerina venezuelana* (Stewart *et al.*, 2012). Our model framework is inherently setup to account for ontogenetic depth changes (we model the intercept, slope and saturation of size- and species-specific isotopes) but there currently is simply too little isotope data available to fit species-specific differences in slope and saturation (mentioned explicitly in the avenues for future research section on lines 161; 387-402).

- **I am also concerned that each of the three grouped time intervals (Late Cretaceous, Paleogene, early Cenozoic, late Cenozoic) contains profound variations in ocean temperature and vertical structure of the water column, but there is no allowance for such complexity in the data analyses, despite the fact that such environmental changes would have strongly influenced interspecies offsets in stable isotopic values.**

As noted in response to Reviewer 1, these colours were for visualization purposes only and not for analysis (bottom of page 6 in this response; now state explicitly in the main text line 124). We apologize for the confusion caused, have stated this “visualization only” clarifier in the revised figure legend and emphasize how we represent the profound differences in environment (the Cramer-Friedrich (2009; 2012) stable isotope curves as continuous explanatory variables) in a revised methods section (lines 142-144). We’ve also replaced Fig. S13 for similar reasons.

- **In my opinion the dataset is too incomplete and assumptions made from it are too simplistic to discern the relative importance of environment, life strategy, phylogeny and preservation in influencing the $\delta^{13}\text{C}$ and $\delta^{18}\text{O}$ values of foraminifera across the past 120 million years.**

We do not claim to have a definitive answer to or the final word on these questions (see caveats section on data limitations, lines 378-402 in the revised text). As the statistician George Box famously remarked, “*all models are wrong, but some models are useful*”. We hope our models are useful in a number of ways. Our key

result is that despite the same methodological treatment to control for environmental, biological and preservational variations, $\delta^{18}\text{O}$ and $\delta^{13}\text{C}$ data infer very different roles for how shared evolutionary history has impacted their expression (lines 295-298). We hope the testing of explicitly evolutionary hypotheses provides a contemporary framework for future studies to test using targeted data collection – one of the major contributions that we envisage this manuscript having is to focus future research efforts on areas of environmental or phylogenetic space where we lack sufficient data at present.

Additional concerns:

- **The text states that the study encompasses data spanning the past 120 m.y. but the oldest data in data table is 106.98 m.y.**

Good spot! The main text has been amended accordingly to read 107 m.y. Thank you.

- **The Cretaceous data set has several time gaps of 3-5 m.y., resulting in incomplete characterization of lineages that span the missing time intervals**

Yes, this is one of the reasons that we limited the phylogenetic analyses to the Cenozoic data (0-65 Ma) where there is both much better data coverage (temporally and phylogenetically) as well as a phylogeny available. However, while patchy, the Cretaceous data still provide a valuable preliminary insight into vital effects in these taxa (see the response to your final main comment).

- **A number of Cretaceous taxa have never been tested for size-related changes in $\delta^{13}\text{C}$ so the assumption that they are asymbiotic may be incorrect**

All of the taxa included in this study have size- $\delta^{13}\text{C}$ and $-\delta^{18}\text{O}$ data available – our data compilation will be published with the paper on the FigShare portal (see private link at <https://figshare.com/s/599d10e3ec1142068ed5>) and draws on five studies: D’Hondt & Zachos 1998 (*Paleobiology*), Houston et al 1999 (*Marine Micropal.*), Houston & Huber 1998 (*Marine Micropaleontology*), Bornemann & Norris 2007 (*Marine Micropal.*), and Norris & Wilson 1997 (*Geology*).

- **An explanation for the data table organization, header abbreviations, numerical depth ecology assignments, preservation rating assignments is missing**

This was previously incorporated into the Supplementary text document (Table S1) and Methods section but to which we now explicitly refer in the Table caption.

- **It is not clear why present water depth is listed when paleowater depth would be more useful**

Good point. Originally, we avoided using paleowater depth due to the varying uncertainty (and source) of paleowater depth estimates: they are generally of low resolution with relatively large uncertainties (± 500 m). However, we have now compiled this list and rerun the analyses with paleowater depth (Pdepth) as an alternative to water depth (watdepth). The code snippet below in red compares the minimum adequate model (MAM) for oxygen with biome removed (o11pdo) to the MAM with biome removed and paleodepth instead of modern water depth (o11pd). We removed biome as the model otherwise failed to converge hindering comparison.

```
o11pdo <- nlme(d180 ~ a + b*meansize + d*meansize^2,
  fixed = list(a~sp+bm+pr+watdepth+depth+cfC+cfO+macro.micro,
    b~sp+watdepth+depth+cfC,
    d~sp+pr+depth+cfO), weights=varIdent(form=~1|fullsp),
  start=rep(0,25), random=a~1|fullsp, data=cenopd[-which(is.na(cenopd$watdepth))])
o11pd <- nlme(d180 ~ a + b*meansize + d*meansize^2,
  fixed = list(a~sp+bm+pr+Pdepth+depth+cfC+cfO+macro.micro,
    b~sp+Pdepth+depth+cfC,
    d~sp+pr+depth+cfO), weights=varIdent(form=~1|fullsp),
  start=rep(0,25), random=a~1|fullsp, data=cenopd[-which(is.na(cenopd$watdepth))])
anova(o11pdo, o11pd)
```

	Model	df	AIC	BIC	logLik
	o11pdo 1	142	5460.652	6343.336	-2588.326
	o11pd 2	142	5528.005	6410.690	-2622.003

The second model, o11pd, using paleodepth has a much higher AIC and therefore a much poorer fit to the data – a difference in AIC scores of 10 is enough for the poorer model to have essentially zero support (p. 71 Burnham & Anderson 2002). As switching from modern water depth to paleodepths explains less variation, and furthermore compromises the stability of the model fits, we have retained the modern depths in the main text. Section 8 in the supplementary appendix contains the above comparison.

- **The sources of data used in the analyses is very heterogeneous; there is no accounting for paleolatitude and local/regional paleoceanographic conditions, which would have influenced relative stable offsets between species (e.g., high clay runoff in nearshore settings would have caused high light attenuation and shallowing of depth habitats; depth habitats would have been influenced by paleolatitude).**

We explicitly included two predictor variables within our initial analyses; biome (either (sub)tropical or polar/transitional), and ocean basin (Atlantic, Pacific or Indian). Although crude, both capture broad trends of the sort highlighted by this comment. Using these factors, we found that geography did a small effect on $\delta^{18}\text{O}$ -size relationships but very little impact on corresponding $\delta^{13}\text{C}$ values (see height of bars in Fig. 2). A more detailed examination of spatial variability is not possible with the data currently available – this is an excellent target for future systematic (spatially and taxonomically complete) sampling and research (suggestion made explicitly on lines 390-402).

- **Some incorrect pore size assignments (e.g., *Biticinella breggiensis* is macroperforate but is listed as finely perforate; *Marginotruncana sinuosa* listed as microperforate but is macroperforate)**

We followed assignments made in the mikrotax database (<http://mikrotax.org/pforams>), which includes the Chronos database as (arguably) the definitive reference for Cretaceous foraminifera. We can't find any peer-reviewed references that contradict our classification, so have, for now, retained these assignments. Please do suggest manuscripts that we may have missed and we shall be happy to update our records.

- **Incorrect genus assignment of *Hedbergella simplex* (should be *Clavihedbergella*)**

Corrected. Thank you.

- **There is no reference list for sources of data in data table**

Thank you for highlighting this oversight, we have now provided the reference list in the Supplement (Section 1).

Reviewer #3 (Remarks to the Author):

- **Edgar et al. test how evolutionary changes may have impacted carbon and oxygen isotope ratios in planktonic foraminifera over the last 120 years. They find that species-specific effects have a large influence on average isotope compositions and on isotopic variability. The authors use statistical methods to assess what predictive factors best describe the histories of $\delta^{13}\text{C}$ and $\delta^{18}\text{O}$ given by planktic foraminifera. The impact of species specific effects is more clearly seen for $\delta^{13}\text{C}$, in line with what is already understood about the importance of vital effects for that isotope system. However, the statistical tests performed in this study more quantitatively demonstrate the importance of considering vital effects in interpreting paleo- $\delta^{13}\text{C}$ from planktic foraminifera. While the importance of foraminiferal growth environment, preservation, size, and vital effects are generally known, this study is significant in that it attempts to tease apart the relative impacts of each. Overall, I think this paper is clearly written and requires minimal revisions. This paper will be important for paleoceanographers and paleoclimatologists who seek to interpret geochemical results derived from planktic forams. In addition, I think this study can serve as a template for investigating other geochemical parameters (Mg/Ca, Sr/Ca, etc.) that are impacted both by environment and biology. The major questions I have are related to the assumption that species-specific effects are only applied to the intercept ('a') and the residuals in equation 1.**

Much appreciated! Please see below for specific responses to the general queries raised above.

- **Introductory paragraph: I found that some of the conclusions stated in the introductory paragraph were a bit inconsistent with the results and conclusions stated later in the paper. In particular, the**

introduction doesn't distinguish between results for $\delta^{18}\text{O}$ and $\delta^{13}\text{C}$ (i.e. line 29-30: "Of the four, the most important is species-specific effects on both the isotopic means and variability"), whereas later in the paper the results clearly imply that species-specific effects are more important for $\delta^{13}\text{C}$ than $\delta^{18}\text{O}$. It would be helpful if the text were clarified in the introductory paragraph.

Done (line 28).

- **Lines 108-110:** The authors state that species-specific effects are only applied to the offset term ('a') and residuals (epsilon) in equation 1. While the authors give general reasons for doing this (lack of sufficient data), some of the results and interpretation seem like they could be strongly influenced by this assumption, giving a circularity (at least as it seemed to me) to some of the major take-homes of the paper. While every model has assumptions underpinning it, I do think it would be useful to more clearly state the implications of this assumption in the main text, and perhaps include some statement regarding how the results might change if vital effects were applied to the other terms as well.

See new text on lines 155-161 and 387-390 that clearly states our assumptions.

- **Line 122:** Perhaps AIC could also be defined in the main text?

Done (line 173), as well as its interpretation as a metric to obtain the most parsimonious description of.

- **Line 132:** I think this conclusion is an interesting one, and if there's sufficient room, it would be nice to explain a little bit further since I didn't find the sentence to be particularly clear (although there is a bit more explanation later in the text). Is the key point here that the expression of species-specific $\delta^{13}\text{C}$ effects depends on how climatic and preservational factors influence the slopes and absolute values? I am also not sure, again, whether this conclusion is a result of the assumption imposed that vital effects are only applied to the offset term 'a'.

No – the 'failure to converge' doesn't hang on including random effects (species-specific effects). It is possible to fit a model including just the fixed effects (the variables shown in Fig. 2) and no species-effects at all (this is fullO in the supporting information), but this model assumes all the data points are independent whereas we know there is substantial dependency in the data: $\delta^{13}\text{C}$ and $\delta^{18}\text{O}$ values are more similar within-species than among-species (see Table 1 for model improvement adding species differences); there is species-specific amounts of variability too (compare fullO with fullOa and fullC with fullCa); and evolutionary history impacts $\delta^{13}\text{C}$ but not $\delta^{18}\text{O}$. We make this point explicit on lines 295-298. The failure to converge is due to the unbalanced nature of the data compilation - the statistical model of $\delta^{13}\text{C}$ does not converge on a most likely solution without preservation and benthic foraminiferal $\delta^{18}\text{O}$, highlighting their overwhelming importance in explaining the variation in the data. We make this point on lines 186-191 and in the new caveats section to guide future data collection efforts.

- **Lines 231 and the last paragraph of the paper:** It would be useful if the authors included another statement about timescale and if they were more explicit about the impacts of turnover for evaluating carbon cycle changes. First, the data are broken up into relatively large chunks of geologic time (10s of millions of years), and the analysis generally implies that the conclusions are relevant over evolutionary timescales. But I wonder how the data would group if you looked at data in finer intervals?

This is an interesting question. There are some limits on how finely we can split the data because of the coverage available, for instance we only have good coverage for 0-2.4 Ma, ~23 Ma, ~39-42 Ma and 56-66 Ma with very little data in the Oligocene, which is a point we now mention explicitly on line 425. As mentioned above, we do not partition time into these three coarse bins and use the colours for visualization only.

- **Second,** there is a statement cautioning that evolutionary changes could obscure environmental signatures at times when turnover is high (mass extinctions, etc.), but it seems like the authors may be able to provide a more concrete estimate for how important this would be during those intervals (i.e. could the variations observed in records be entirely due to evolutionary shifts)?

This is a longer term goal of this work but at this point we caution against directly applying the values calculated here to 'correct' paleoceanographic records because of structural limitations in the datasets (lines 378-380).

Reviewer #4 (Remarks to the Author):

- **This manuscript explores the potential for evolutionary history to affect foram isotopic records over the last 120 million years. The authors find a strong phylogenetic signal for carbon isotope vital effects, but none for oxygen isotopes. This is at least partially reassuring for those who use these isotopes. The manuscript is well written.**

Thanks!

- **The ideal test for vital effects would compare the isotopic composition of the foram tests with the water they grew in. This is obviously only available for extant species (and may be difficult even for these). For the fossil forams, data on the isotopic composition of the water are not available, and instead the authors have to rely upon the biogeographic affiliation and a benthic foram oxygen isotope derived estimate of global climate. This probably limits the potential to find a phylogenetic signal in the oxygen isotopes.**

This is entirely possible, and we state this reasoning on lines 284-287 in our revision. However, oxygen is also much less heavily fractionated than carbon by biological processes in both foraminifera and other many other marine calcifiers (see summary in Norris, 1998 but also McConaughy, 1989, Wefer and Berger, 1991; Stanley and Swart, 1995) meaning that any offsets associated with oxygen (if present) are likely to be much smaller making them harder to detect but fortunately much less of a problem for environmental reconstructions too. See response below for more on this.

- **In their model, the authors include foram characteristics such as wall morphology as predictors. As these characteristics are conserved, these predictors probably explain some of the variance that could also be explained by phylogenetics. This could also explain why there is no phylogenetic signal in the oxygen isotopes. Whether it is useful to include these predictors depends on the question being asked. If the authors are interested in the full phylogenetic signal, they should probably be excluded. If they are interested in the residual phylogenetic signal after obvious biological predictors are accounted for, they should be included. For applied work, the second is probably more useful. This could be explored.**

Excellent suggestion – the biological and ecological factors included in the analysis (symbionts, spinosity, wall structure, and depth habitat) all have the potential to reduce our ability to detect a phylogenetic signal in the carbon and oxygen data because these very factors also have a strong evolutionary signal. Adopting this suggestion to remove the biological effects from oxygen, then we retain our maximum likelihood estimate of λ at 0 (see supplementary material section 7 and line 284-290 in the main manuscript). There are two ways of doing this: if we remove the biological effects from the nls stage but keep them in the pgl's stage, then the 95% CI is (NA, 0.455) - up from (NA, 0.405); if we remove the biological effects from both nls and pgl's then the 95% CI is (NA, 0.385) - down from (NA, 0.485). The analogous models for Carbon did not converge.

Our intention in controlling for the biological and environmental effects was motivated by accounting for the total variance in the isotopic data using all the readily available biological, ecological and environmental data we could find for the entire dataset. The similarity in λ estimates with and without these variables, and their weak or non-significant impacts in the pgl's regression (supplement section 7) suggests that the dependence among species is more driven by their shared evolutionary history than shared life-histories: two closely related symbiont bearing species will, according to our predictions, be more similar to each other than two distantly related symbiont bearing species, for example.

Taking up the Reviewer's suggestion that "this could be explored", we discuss both sets of results in the revised manuscript (lines 284-290). After stating the results, we make the point that the test of a phylogenetic signal in the residual variance is really key. A number of checks and approaches still need to be investigated to provide robust offsets to the community from an applied analytical perspective (see caveats section line 320).

- **It is not immediately obvious why the authors are using a non-linear mixed effects model when there are no nonlinear terms in their model - all the terms, including the quadratic - are additive. This could be explained.**

The non-linearity is taken from the section headers in the Pinheiro & Bates (2000) textbook (from page 271), which we now reference at this point in the text. The non-linearity is in the assumed functional form for the size-dependence and not the additive statistical phase.

- **It would be useful to discuss the magnitude of the vital effect revealed by the analysis (whether there is a phylogenetic signal or not), and how large it is relative to the apparent signal in isotopic records.**

Excellent point. We can approximate this by calculating the ratio of the random effect to the smallest projected value for $\delta^{13}\text{C}$ and $\delta^{18}\text{O}$ as appropriate. There are two huge caveats here: as the smallest projected value tends to zero, so the ratio heads to infinity; and these ratios reflect the very unbalanced data compilation we have pulled together and not general expectations. These caveats notwithstanding, the numbers are 0.459 for $\delta^{13}\text{C}$ and 0.714 for $\delta^{18}\text{O}$. The calculations emphasize that we are not arguing that vital effects are unimportant for $\delta^{18}\text{O}$, merely that the random effects are systematic for $\delta^{13}\text{C}$ and not $\delta^{18}\text{O}$. We are not comfortable presenting the figures because the random effects will also contain the local environmental variation of each site where a species is sampled and we cannot distinguish, under the current sampling design, between abiotic variation and the species-specific offset, but include this general take-home message in the Discussion (lines 379-402). This would be a brilliant project for future study.

- **I don't really understand figure 3. I think it is the phrase "available size data" which is confusing me as all the data included had to have size data.**

See response to reviewer 1 (Helen Coxall) above (page 6 of this letter) – we used “available size data” to refer to the size range of data available for the species in question. For instance, if a species only had data available from 100-300 microns, we only projected its values over 100-300 microns to avoid extrapolation. We hope the revised wording is unambiguous.

- **I greatly appreciate the authors providing the Rmarkdown output with all the scripts. I would recommend, however, that this code and the data are archived, for example in a github repository, rather than making it available from the authors by request.**

No problem – we have to set up a figshare folder with everything in it from this paper and its precursor (Ezard *et al.* 2015). <https://figshare.com/s/599d10e3ec1142068ed5> is currently a private link but contains the knitr (similar to Rmarkdown), compiled pdfs and data files used. Although github has versioning, it would be harder to extract the actual code used for analyses without a “version of record” that remains static. If there is anything else you would appreciate, let us know and we'll add it to this fileset before publishing it with a DOI.

References

1. Bornemann, A. & Norris, R. D. Size-related stable isotope changes in Late Cretaceous planktic foraminifera: Implications for paleoecology and photosymbiosis. *Marine Micropaleontology* **65**, 32-42, doi:<http://dx.doi.org/10.1016/j.marmicro.2007.05.005> (2007).
2. Burnham, K. P., and D. R. Anderson. "Information and likelihood theory: a basis for model selection and inference." *Model selection and multimodel inference: a practical information-theoretic approach 2* (2002): 49-97.
3. Cooper, N., Thomas, G. H. and FitzJohn, R. G. (2016), Shedding light on the 'dark side' of phylogenetic comparative methods. *Methods Ecol Evol*, 7: 693–699. doi:10.1111/2041-210X.12533
4. Cramer, B. S., Toggweiler, J. R., Wright, J. D., Katz, M. E. & Miller, K. G. Ocean overturning since the Late Cretaceous: Inferences from a new benthic foraminiferal isotope compilation. *Paleoceanography* **24**, PA4216, doi:10.1029/2008pa001683 (2009).
5. D'Hondt, S. & Zachos, J.C. Cretaceous foraminifera and the evolutionary history of planktic photosymbiosis. *Paleobiology* **24**(04), 512-523 (1998).
6. Edgar, K.M., Anagnostou, E., Pearson, P.N., and Foster, G.L. Assessing the impact of diagenesis on $\delta^{11}\text{B}$, $\delta^{13}\text{C}$, $\delta^{18}\text{O}$, Sr/Ca and B/Ca values. *Geochimica et Cosmochimica Acta* **166**, 189-209.
7. Friedrich, O., Norris, R. D. & Erbacher, J. Evolution of middle to Late Cretaceous oceans—A 55 m.y. record of Earth's temperature and carbon cycle. *Geology* **40**, 107-110, doi:10.1130/g32701.1 (2012).

8. Houston, R. M. & T. Huber, B. Evidence of photosymbiosis in fossil taxa? Ontogenetic stable isotope trends in some Late Cretaceous planktonic foraminifera. *Marine Micropaleontology* **34**, 29-46, doi:[http://dx.doi.org/10.1016/S0377-8398\(97\)00038-8](http://dx.doi.org/10.1016/S0377-8398(97)00038-8) (1998).
9. Houston, R. M., Huber, B. T. & Spero, H. J. Size-related isotopic trends in some Maastrichtian planktic foraminifera: methodological comparisons, intraspecific variability, and evidence for photosymbiosis. *Marine Micropaleontology* **36**, 169-188 (1999).
10. Norris, R. D. & Wilson, P. A. Low-latitude sea-surface temperatures for the mid-Cretaceous and the evolution of planktic foraminifera. *Geology* **26**, 823-826, doi:10.1130/0091-7613(1998)026<0823:llsstf>2.3.co;2 (1998).
11. Stewart, J. A., Wilson, P. A., Edgar, K. M., Anand, P. & James, R. H. Geochemical assessment of the palaeoecology, ontogeny, morphotypic variability and palaeoceanographic utility of "*Dentoglobigerina*" *venezuelana*. *Mar Micropaleontol* **84-85**, 74-86, doi:Doi 10.1016/J.Marmicro.2011.11.003 (2012).

REVIEWERS' COMMENTS:

Reviewer #1 (Remarks to the Author):

Comments on the revised ms by Helen Coxall

The authors have put a lot of effort into addressing the reviewer's comments and the message and implications are thus clearer and more accessible, however a little further tweaking would help to drive home the purpose of this effort, i.e. link users (palaeoceanographers) with their methods/toolbox and the broader science goals.

It will be interesting to see how exactly this will be applied to a future long term planktic isotope curve for example.

A few final comments, below and in the attached PDF with added comments.

The new title works.

Line 21: I think this is where you have to emphasize the very long time frames that you are considering here, i.e. its really only long geological time scales where large evolutionary jumps are occurring where evolutionary histories become an issue. By ironing out this point early on I think everything becomes clearer.

Lines 28- "Our results lay the groundwork for a phylogenetic approach for bias-free inference from long-term ^{13}C records – a key measure of holistic organismal biology and of the global carbon cycle."

This sentence is getting close to the essence but I find it still doesn't say much; i.e. "

I can see the authors have worked a lot with the text and the new clarifications help a lot. There are sections in the rebutal to my comments that I think however are even clearer: e.g.

"At present, paleoceanographers and geochemists acknowledge that vital effects are important and state they account for them through symbiont effects, depth habitat, and measuring isotopic offsets between specific species. Closely related species within the same genus are chosen for long-term environmental reconstructions and then across extinction boundaries or between low-and-high latitude assemblages we jump to other clades on the basis of symbiont ecology, depth habitat, and relative isotopic differences. Geochemists therefore use an implicit evolutionary hypothesis, but do not test it formally. We seek to move beyond this ad-hoc incorporation of vital effect dependence to an explicitly evolutionary setting embedding changes among species and correlating biological effects in mathematical models of evolutionary divergence. "

I really like this explanation, linking 'users' with proxy and scientific goal, and I think others will equally find some formulation of this very helpful and because it links users to method improvements via theory.

Line 96: You probably want to make the point that a planktonic foram compilation is a very different thing to the benthic Cenozoic curve (which many are now familiar with); across 70 million years, there are jumps between different benthic taxa (needed to get the full time coverage) accommodating evolution but that is probably easier to get 'right' for benthics because we have slightly more confidence that the benthic environment and thus benthic ecologies are more conservative in the first place.

Re the benthic foram $\delta^{18}\text{O}$ 'environmental predictor. This is described as representing "high latitude temperatures. However, this assumes that both northern and southern high latitudes produced deep waters, whereas this is quite uncertain, especially for pre Oligocene time. I suggest

restricting this to 'southern high latitudes', since proxies suggest deep water was forming around Antarctica throughout the Cenozoic. Fix in the fig caption. Too.

It seems that Figure 3 is referred to before figure 2.

Is the Fig 2 callout missing or is the order wrong?

Interpreting Fig 2: So what's the relevance of a long/tall bar for the interpretation?

Lines 216-220: I think the point here is that this becomes very relevant when you start looking at the long-term (geological time scales) ocean histories. There hasn't been a planktic, i.e. upper ocean 'Zachos curve'. But if/as we start to build this THEN your findings DO become really important. But no one has/had really done it yet. I think you need to make this all clear.

Reviewer #2 (Remarks to the Author):

The authors have done a fabulous job addressing reviewer questions and concerns by re-analyzing some of their data, inserting revisions to the text to improve clarity and to add insight, revising and adding figures, and providing justification for not making some suggested changes. Their assessment of how phylogenetic constraints and vital effects influence $\delta^{13}\text{C}$ and $\delta^{18}\text{O}$ signatures for planktic foraminiferal taxa and their generation of predictive models have important significance to the paleoceanographic and paleoclimatic research communities and will provide a good basis for future studies. I now consider the manuscript worthy of publication in Nature Communications. The revised text is finely written and all illustrations are presented clearly and are necessary for the manuscript. I recommend acceptance without further revision.

Reviewer #3 (Remarks to the Author):

Edgar et al. have made thorough and complete edits to their original submission and it appears that points raised by previous reviewers have been satisfactorily addressed. As stated previously, this manuscript is of importance for the paleoceanographic community because of the authors' ability to quantitatively demonstrate how important $\delta^{13}\text{C}$ vital effects can be for interpreting carbon isotope records. I was impressed by the comprehensiveness of the authors' supplementary materials, and I think those materials will offer a great deal of transparency for the statistical analyses involved. Responses to previous reviewer comments were thoughtful and in many cases the conclusions and major points of the main text are more clearly stated in the revised version, which is helpful. I believe this manuscript should be accepted to Nature Communications.

Reviewer #4 (Remarks to the Author):

I only have minor comments on this version of the manuscript.

1) I remain unpersuaded that the authors use a non-linear model. The model used

```
nullO <- nlme(d18O ~ a + b*meansize + d*meansize^2,  
fixed=list(a~1, b~1, d~1),  
random=a~1|fullsp,  
data=ceno, start=out)
```

There is no non-linear term in this model. The squared term does not make the model non-linear.

This model can be rewritten as a linear mixed effects model as

```
nullO <- lme(d18O ~ meansize + I(meansize^2),
```

```
random = 1|fullsp,  
data = ceno, start = out)
```

While this will make little difference to the results (both should converge to the same solution), but should clarify the modelling approach used and might reduce problems with model convergence.

2) The PDF format for Supplementary Datafile 1 is not ideal. Extracting data from a pdf is tedious. A csv file would be much more useful, uploaded to figshare if necessary.

3) The SI suggests that the author contact the author for markdown file rather than figshare.

4) Consider using GGally::ggpairs() for figure s1 as it should give a cleaner plot than the base graphics currently used.

Response to 2nd round of Reviewer's Comments for manuscript "Evolutionary history biases inferences of ecology and environment from $\delta^{13}\text{C}$ but not $\delta^{18}\text{O}$ values"

For ease, we have included the original comments from each of the reviewers that required an action in bold (comments transferred across from the annotated PDF are shown in bold italics) and our response to comments beneath in blue.

Reviewer #1: Comments on the revised ms by Helen Coxall

The authors have put a lot of effort into addressing the reviewer's comments and the message and implications are thus clearer and more accessible, however a little further tweaking would help to drive home the purpose of this effort, i.e. link users (palaeoceanographers) with their methods/toolbox and the broader science goals. It will be interesting to see how exactly this will be applied to a future long term planktic isotope curve for example. A few final comments, below and in the attached PDF with added comments.

The new title works.

Thank you!

Line 21: I think this is where you have to emphasize the very long time frames that you are considering here, i.e. its really only long geological time scales where large evolutionary jumps are occurring where evolutionary histories become an issue. By ironing out this point early on I think everything becomes clearer.

Whilst comparing $\delta^{13}\text{C}$ records between time periods with different levels of relatedness is perhaps the most obvious application of this work, this is not the only situation where evolutionary relatedness is relevant. For instance, it could help to explain unexplained $\delta^{13}\text{C}$ differences between species inferred to live at the same depth habitats at a single location in a single timeslice, between taxa from different biomes within a single timeslice or across habitats but through time. Thus, we do not reference a specific timescale but rather modify the final sentence to make it clear that our approach is more widely applicable. This should also come through in the revised introductory paragraph L58-76.

Lines 28- "Our results lay the groundwork for a phylogenetic approach for bias-free inference from long-term ^{13}C records – a key measure of holistic organismal biology and of the global carbon cycle." This sentence is getting close to the essence but I find it still doesn't say much; i.e. "

We have edited this final sentence (L30) to try and make it clearer that our work ultimately aims to develop a means of 'correcting' for modern and extinct planktonic foraminiferal vital effects on $\delta^{13}\text{C}$ records through time, providing a mechanistic understanding of how "vital effect" differences are generated, to facilitate more accurate reconstructions of organismal ecology and the global carbon cycle through time.

I can see the authors have worked a lot with the text and the new clarifications help a lot. There are sections in the rebuttal to my comments that I think however are even clearer: e.g.

"At present, paleoceanographers and geochemists acknowledge that vital effects are important and state they account for them through symbiont effects, depth habitat, and measuring isotopic offsets between specific species. Closely related species within the same genus are chosen for long-term environmental reconstructions and then across

extinction boundaries or between low-and-high latitude assemblages we jump to other clades on the basis of symbiont ecology, depth habitat, and relative isotopic differences. Geochemists therefore use an implicit evolutionary hypothesis, but do not test it formally. We seek to move beyond this ad-hoc incorporation of vital effect dependence to an explicitly evolutionary setting embedding changes among species and correlating biological effects in mathematical models of evolutionary divergence. "

I really like this explanation, linking 'users' with proxy and scientific goal, and I think others will equally find some formulation of this very helpful and because it links users to method improvements via theory. Can you sharpen these introductory ideas with a closer match to those words?

Yes, absolutely – see revised section on L58-76.

L78 - clarify that this is a phylogeny based on (arguably) comparative morphology in the first place, and thus morphospecies,...so some circularity..?

Done (L448-451).

L81 - again, what do you mean by life history? Example?

Life history is the series of changes that an organism undergoes during its lifetime (in essence its life cycle) and includes (but is not limited to) biological processes influencing survival, growth and reproduction, e.g., diet, body size, ontogenetic depth habitat changes. We have added clarification on L86.

Line 96: You probably want to make the point that a planktonic foram compilation is a very different thing to the benthic Cenozoic curve (which many are now familiar with); across 70 million years, there are jumps between different benthic taxa (needed to get the full time coverage) accommodating evolution but that is probably easier to get 'right' for benthics because we have slightly more confidence that the benthic environment and thus benthic ecologies are more conservative in the first place.

This is a good point, long multi-million year benthic foraminiferal records are likely less susceptible to offsets from accumulated evolutionary differences than their planktic counterparts (assuming no within species changes through time), which emphasizes the importance of understanding how evolutionary history impacts $\delta^{13}\text{C}$. We defer this point to the Discussion (L420-425) to maintain the focus in the introduction and until after the necessary background information on the evolutionary models are presented.

L132 - there seems to be an error in this sentence

Sentence now edited for clarity (L140-142).

Lines 216-220: I think the point here is that this becomes very relevant when you start looking at the long-term (geological time scales) ocean histories. There hasn't been a planktic, i.e. upper ocean 'Zachos curve'. But if/as we start to build this THEN your findings DO become really important. But no one has/had really done it yet. I think you need to make this all clear.

See comment on page 1 – phylogenetic biases matter both through time and among contemporaneous species. We edited the introductory paragraph (L58-76) to highlight this key point.

L327 - Re. the benthic foram d180 'environmental predictor. This is described as representing "high latitude temperatures. However, this assumes that both northern and southern high latitudes produced deep waters, whereas this is quite uncertain, especially for pre Oligocene time. I suggest restricting this to 'southern high latitudes', since proxies suggest deep water was forming around Antarctica throughout the Cenozoic. Fix in the fig caption.

The North Atlantic has been a key region for deep-water formation since the middle Eocene (~42 Myrs at least) encompassing a good portion of our dataset. Prior to this there is empirical and model evidence (albeit limited) for northern sourced deep water in the Pacific in the early Paleogene and possibly in the late Cretaceous (e.g., Thomas et al., 2008; Hague et al., 2012; Donnadieu et al., 2015). Given this uncertainty and that we know that North Atlantic is an important source of deep water for the past ~42 Myrs we feel 'high latitude' is appropriate but have added a caveat in the Methods to address the reviewers concern (L468-474).

It seems that Figure 3 is referred to before figure 2. Is the Fig 2 callout missing or is the order wrong?

The call out for Figure 2 first appears on L193 and for Figure 3 on L208.

Interpreting Fig 2: So what's the relevance of a long/tall bar for the interpretation?

The height of each bar indicates the explanatory power of each of the tested environmental, preservational and ecological variables: the higher the bar, the greater the explanatory power of a particular variable on a component (intercept, linear or non-linear) of the size-isotope relationship. This is fully described in Figure Caption 2 but we now also include a brief mention in the main text on L193.

Figure 2 - Why are some bars longer than the scale (0-100)?

Several bars are extended beyond the scale to clearly highlight the importance of these variables for explaining size-isotope trends, but including their full limits (up to 461.9) would obscure the other relevant drivers, hence the break.

Figure 5 - Since you use S. velascoensis in this case study here, can you also map it or all subbotinids onto the previous figure as done for Truncatulinoidea. This should show for starts that this clades occupies a different sector of isotope space.

Good suggestion. We have now added a third panel to Figure 4 (now 4b) showing the relative clumping of the Paleogene subbotinids to help continuity to Fig. 5.

L333 - A forth potential factor might be varying associations with bacterial planktonic foram symbionts under differing climate modes - this seems to be something new coming out of studies of modern foraminifera.

Absolutely. We initially missed this reference (Bird et al., 2017) but now include bacterial symbiosis as a possible fourth option to explain the importance of background climate state indicated by benthic foraminiferal $\delta^{18}\text{O}$ on size- $\delta^{13}\text{C}$ trends (L345).

Reviewer #2 (Remarks to the Author):

The authors have done a fabulous job addressing reviewer questions and concerns by re-analyzing some of their data, inserting revisions to the text to improve clarity and to add insight, revising and adding figures, and providing justification for not making some suggested changes. Their assessment of how phylogenetic constraints and vital effects influence d13C and d18O signatures for planktic foraminiferal taxa and their

generation of predictive models have important significance to the paleoceanographic and paleoclimatic research communities and will provide a good basis for future studies. I now consider the manuscript worthy of publication in Nature Communications. The revised text is finely written and all illustrations are presented clearly and are necessary for the manuscript. I recommend acceptance without further revision.

Thank you - much appreciated!

Reviewer #3 (Remarks to the Author):

Edgar et al. have made thorough and complete edits to their original submission and it appears that points raised by previous reviewers have been satisfactorily addressed. As stated previously, this manuscript is of importance for the paleoceanographic community because of the authors' ability to quantitatively demonstrate how important $\delta^{13}\text{C}$ vital effects can be for interpreting carbon isotope records. I was impressed by the comprehensiveness of the authors' supplementary materials, and I think those materials will offer a great deal of transparency for the statistical analyses involved. Responses to previous reviewer comments were thoughtful and in many cases the conclusions and major points of the main text are more clearly stated in the revised version, which is helpful. I believe this manuscript should be accepted to Nature Communications.

Thank you - much appreciated!

Reviewer #4 (Remarks to the Author):

I only have minor comments on this version of the manuscript.

1) I remain unpersuaded that the authors use a non-linear model. The model used

```
null0 <- nlme(d180 ~ a + b*meansize + d*meansize^2,  
fixed=list(a~1, b~1, d~1),  
random=a~1|fullsp,  
data=ceno, start=out)
```

There is no non-linear term in this model. The squared term does not make the model non-linear.

This model can be rewritten as a linear mixed effects model as

```
null0 <- lme(d180 ~ meansize + I(meansize^2),  
random = 1|fullsp,  
data = ceno, start = out)
```

While this will make little difference to the results (both should converge to the same solution), but should clarify the modelling approach used and might reduce problems with model convergence.

That is only correct for the null model without explanatory variables – the main body of the manuscript uses explanatory variables (Fig. 2), including the model averaging approach, to answer the titular result that $\delta^{13}\text{C}$ is biased by evolutionary history. Without the assumed non-linear structure of the explanatory variables impacting size (e.g., warmer ambient temperature

impacts organismal size through the a, b and c coefficients) we could not construct a meaningful synthetic column to allow us to contrast all species in a comparable data format.

2) The PDF format for Supplementary Datafile 1 is not ideal. Extracting data from a pdf is tedious. A csv file would be much more useful, uploaded to figshare if necessary.

Supplementary Datafile 1 was initially uploaded as a PDF for ease of review but we have now included the .csv file instead to permit easier reuse.

3) The SI suggests that the author contact the author for markdown file rather than figshare.

Thank you – changed. The DOI is now provided in the main manuscript and the SI (10.6084/m9.figshare.5048854, not yet active because we do not have final acceptance – see the current status via <https://figshare.com/s/599d10e3ec1142068ed5>).

4) Consider using GGally::ggpairs() for figure s1 as it should give a cleaner plot than the base graphics currently used.

Possibly, but the residuals plot generated uses an internal nlme function not base:pairs() so the arguments to be passed to the functions are different. Our goal with this figure is to make our analytical decisions transparent, which we hope will, in turn, encourage others to adopt an holistic approach to open science. Figure S1 achieves this using one line of code.